# A Systematic Evaluation of Domain Adaptation Algorithms on Time Series Data

## Abstract

Unsupervised domain adaptation methods aim to generalize well on unlabeled test data that may have a different (shifted) distribution from the training data. Such methods are typically developed on image data, and their application to time series data is less explored. Existing works on time series domain adaptation suffer from inconsistencies in evaluation schemes, datasets, and backbone neural network architectures. Moreover, labeled target data are usually employed for model selection, which violates the fundamental assumption of unsupervised domain adaptation. To address these issues, we propose a benchmarking evaluation suite (AdaTime) to systematically and fairly evaluate different domain adaptation methods on time series data. Specifically, we standardize the backbone neural network architectures and benchmarking datasets, while also exploring more realistic model selection approaches that can work with no labeled data or just few labeled samples. Our evaluation includes adapting state-of-the-art visual domain adaptation methods to time series data in addition to the recent methods specifically developed for time series data. We conduct extensive experiments to evaluate 10 state-of-the-art methods on 4 representative datasets spanning 20 cross-domain scenarios. Our results suggest that with careful selection of hyper-parameters, visual domain adaptation methods are competitive with methods proposed for time series domain adaptation. In addition, we find that hyper-parameters could be selected based on realistic model selection approaches without relying on labeled samples from target domain. Our work unveils practical insights for applying domain adaptation methods on time series data, and builds a solid foundation for future works in the field.

## 1 Introduction

Time series classification problem is predominant in many real-world applications including healthcare and manufacturing. Yet, time series data with its inherit temporal dynamics and complex underlying pattern can be more challenging than static data. Recently, deep learning has gained more attention in time series classification tasks, assuming access to a vast amount of labeled data for training (Fawaz et al., 2019). However, annotating time series data can be challenging and burdensome due to its complex underlying patterns and complicated domain knowledge (Chang et al., 2020). One way to reduce labeling burden is to leverage annotated data (e.g., synthetic or public data) from a relevant domain (i.e., source domain) to train the model, while testing it on the domain of interest (i.e., target domain). Nevertheless, deep learning tends to perform poorly when tested on unseen data that have different distribution from the training data, which is well-known as the domain shift problem. Considering time series applications, source and target domains usually represent data from different subjects (persons) as in human activity recognition (HAR) (Wilson et al., 2020; Chang et al., 2020) or sleep stage classification (SSC) tasks (Phan et al., 2020). A considerable amount of literature has been developed on Unsupervised Domain Adaptation (UDA) for visual applications to mitigate the domain shift problem (Long et al., 2015; Ganin et al., 2016; Long et al., 2018; Chen et al., 2020; Rahman et al., 2020).

Recently, more attention has been paid to time series UDA (TS-UDA) (Wilson et al., 2020; Chang et al., 2020; Ragab et al., 2020; Liu & Xue, 2021). However, the literature of TS-UDA methods suffers from the following limitations:

- Existing TS-UDA works lack consistent evaluation schemes including benchmark datasets, preprocessing, and backbone networks. For instance, some methods that leverage recurrent neural network as backbone network (Purushotham et al., 2017) are compared against methods with convolutional based backbone networks (Wilson et al., 2020). In addition, even with a similar architecture, training procedures can also vary among different algorithms including number of epochs, weight decay, and learning rate schedulers (Tonutti et al., 2019; Purushotham et al., 2017).

- Existing TS-UDA works use labeled data from the target domain for model selection, violating the basic assumption of UDA (Wilson et al., 2020; Liu & Xue, 2021), and providing an overoptimistic view of their performance. It is worth noting that model selection for domain adaptation in the absence of target domain labels is a long standing problem.

- Most of existing algorithms are mainly *application specific*, and few works have been proposed for general TS-UDA. As a result, there is a shortage of baseline methods when applying domain adaptation on time series data.

All the aforementioned challenges can highly affect the performance and can be mistakenly attributed to the proposed domain adaptation methods.

In this work, we propose a systematic evaluation suite (ADATIME) to tackle the aforementioned obstacles and remove all extraneous factors to ensure a fair evaluation of different UDA algorithms on time series data. Specifically, to address the inconsistent evaluation schemes and backbone networks, we first standardize the preparation and processing of four benchmarking datasets from two classic real-world applications, including healthcare and human activity recognition. Besides, we unify the backbone network and the training procedures when comparing between different UDA methods. Second, to select the model hyper-parameters in the absence of target labels, we explore more realistic model selection strategies for TS-UDA problem that do not require target labels. Particularly, we investigate the model performance when selecting models based on source dataset (i.e., source risk (Ganin et al., 2016)), unlabeled target data (i.e., deep embedded evaluation (DEV) (You et al., 2019)), or only few-shot labeled samples from target data. Last, to address the lack of TS-UDA baselines, we re-implement various state-of-the-art visual UDA methods that can be adapted to time series data while comparing to the existing TS-UDA methods

Given our standard methodology, we aim to systematically study the following questions: (1) With standard backbone network and evaluation schemes, how will the visual UDA methods perform on time series data; (2) Can we use realistic model selection methods —relying on only few or no target labels— and still achieve acceptable adaptation performance on time series data; (3) How can the backbone network contribute to the performance. In this paper, we conduct comprehensive experiments to answer the aforementioned three questions. Some of our findings are summarized as follows:

- Visual domain adaptation methods can achieve comparable or even better performance than the methods proposed for time series data.

- Unlike image data, selecting models based on source risk can achieve reasonable performance and outperforms the DEV risk for time series data. Additionally, we find that our proposed few-shot target risk can achieve comparable performance to target risk, with affordable few labels samples.

- Changing the base architecture can be pivotal to the performance. Moreover, complex architectures with time series data can have lower generalization performance on cross-domain experiments.

## 2 DOMAIN ADAPTATION PROBLEM

We start by defining the unsupervised domain adaptation problem. We assume the access to a labeled source domain $X_S = \{(\mathbf{x}_i^s, y_i^s)\}_{i=1}^{N_S}$ that represents uni-variate or multivariate time series data, and unlabeled target domain $X_T = \{(\mathbf{x}_i^T)\}_{i=1}^{N_T}$ where $N_S$ and $N_T$ denote the number of samples for $X_S$ and $X_T$ respectively. The source and target domains are sampled from different marginal distributions, i.e., $P_S(x) \neq P_T(x)$, while the conditional distribution remains stable ($P_S(y|x) =$

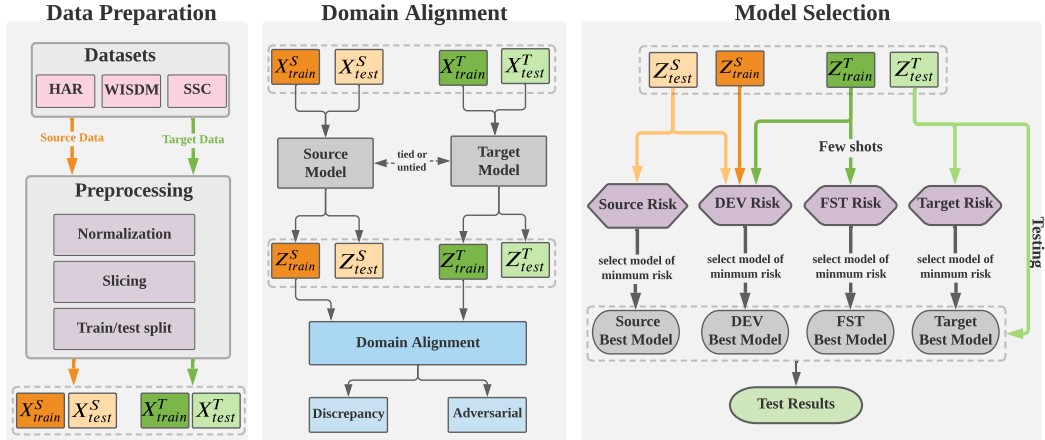

Figure 1: Our benchmarking methodology ADATIME consists of three main steps: Data Preparation, Domain Alignment, and Model Selection. We first prepare the train and test data in both source and target domains (i.e., $X_{train}^S, X_{test}^S, X_{train}^T, X_{test}^T$). Then the processed source and target data are passed through the backbone network to extract the corresponding features. The domain alignment algorithm being evaluated is then used to address the distribution shift between the source and target features. Last, we pass the source and target data to select the best hyper-parameters for the domain alignment algorithm. We used three different model selection approaches (Best viewed in color).

$P_T(y|x)$). The main goal of UDA is to reduce the distribution shift between $P_S(x)$ and $P_T(x)$, assuming they share the same label space.

The mainstream of UDA algorithms is addressing the domain shift problem by finding domain invariant feature representation. Formally, given a backbone model $f_\theta : X \to Z$, which transforms the input space to the feature space, the UDA algorithm mainly optimizes the backbone network to find new invariant representation for the target features such that $P_s(f_\theta(x)) = P_t(f_\theta(x))$. As a results, a model trained on the source domain can generalize well to the adapted target domain.

## 3 ADATIME: A BENCHMARKING APPROACH FOR TIME SERIES DOMAIN ADAPTATION

In this work, we systematically evaluate domain adaptation algorithms on time series data, ensuring fair and realistic procedures. Figure 1 shows the details of ADATIME, which proceeds as follows. Given a dataset, we first apply our standard time series preparation schemes on the selected source and target domains. Subsequently, the source train data $X_{train}^S$ and target train data $X_{train}^T$ are passed through the backbone network to extract source and target features, i.e., $Z_{train}^S$ and $Z_{train}^T$ respectively. The selected UDA algorithm is then applied to mitigate the distribution shift between the extracted features of two domains. Last, to set the hyper-parameters of the UDA algorithm, we consider three practical model selection approaches (i.e., without any target domain labels or only few-shot labeled samples) namely source (SRC) risk, deep embedded evaluation (DEV) risk, and few-shot target (FST) risk. The proposed evaluation pipeline can prevent any extraneous factors and enable fair comparison between different UDA methods. The code of ADATIME will be made publicly available for researchers to enable seamless evaluation of different domain adaptation methods on time series data.

### 3.1 BACKBONE NETWORK

The general domain adaptation network consists of a feature extractor, a classifier, as well as different components to align the domains. We refer to the feature extractor and the classifier as the backbone network. The backbone network can have a pivotal contribution to the UDA model performance, as using different architectures may result in different performance. Hence, changing the backbone network when comparing different UDA methods can result in misleading outcomes,

which can hinder the fair evaluation protocol. In most visual applications, ResNet-50 is the dominantly adopted feature extractor, as its complex architecture can extract representative features from images. However, in time series, using such complex architectures can lead to overfitting problems (Fawaz et al., 2019). Therefore, we use a 1D-CNN architecture in ADATIME (see Figure 6). In addition, we study the effect of different backbone architectures on the performance of TS-UDA methods.

## 3.2 DOMAIN ADAPTATION ALGORITHMS

While numerous UDA approaches have been proposed to address the domain shift problem (Zhao et al., 2020b), a comprehensive review of existing UDA methods is out of our scope. Instead, we only included the most solid and recent baselines for visual UDA that can be adapted to time series. Besides, we included the state-of-the-art methods proposed for time series data. Overall, the selected algorithms in ADATIME can fit into two main categories according to the domain adaptation method, namely discrepancy based methods and adversarial based methods. The former aims to minimize a statistical distance between source and target features to mitigate the domain shift problem (Tzeng et al., 2014; Sun & Saenko, 2016; Chen et al., 2020). The latter leverages a domain discriminator network that enforces the feature extractor to produce domain invariant features (Ganin et al., 2016; Tzeng et al., 2017). Each method in the above categories can also be classified according to the aligned distribution. Specifically, some algorithms only align the marginal distribution of the feature space, while other algorithms jointly align the marginal and conditional distributions.

The selected UDA algorithms are as follows: Deep Domain Confusion (**DDC** Tzeng et al. (2014)); Higher-order Moment Matching (**HoMM** Zhao et al. (2020a)); Correlation Alignment via Deep Neural Networks (**Deep-CORAL** Sun et al. (2017)); Minimum Discrepancy Estimation (**MMDA** Rahman et al. (2020)); Domain-Adversarial Training of Neural Networks (**DANN** Ganin et al. (2016)); Conditional Adversarial Domain Adaptation (**CDAN** Long et al. (2018)); Deep Subdomain Adaptation (**DSAN** Zhu et al. (2021)); (**DIRT-T** Shu et al. (2018)). In addition, we also include two UDA methods applied in time series classification, which are **CoDATS** (Wilson et al., 2020) and **AdvSKM** (Liu & Xue, 2021), noting that there are very few TS-UDA methods in the literature. We excluded the methods proposed for time series prediction/forecasting since they are out of our scope. Table 1 shows a summary of the selected methods and their corresponding categories.

Table 1: Summary of Domain Adaptation algorithms used for our benchmarking framework

| Algorithm | Application | Category | Distribution | Losses | Model Selection |
|---|---|---|---|---|---|
| DDC | Visual | Discrepancy | Marginal | MMD | Target Risk |
| Deep-Coral | Visual | Discrepancy | Marginal | CORAL | Not Mentioned |
| HoMM | Visual | Discrepancy | Marginal | High-order MMD | Not Mentioned |
| MMDA | Visual | Discrepancy | Joint | MMD, CORAL, Entropy | Target Risk |
| DSAN | Visual | Discrepancy | Joint | Local MMD | Not Mentioned |
| DANN | Visual | Adversarial | Marginal | Domain Classifier, Gradient Reversal Layer | Source Risk |
| CDAN | Visual | Adversarial | Joint | Conditional adversarial Domain Classifier | Importance Weighting |
| DIRT-T | Visual | Adversarial | Joint | Virtual adversarial Entropy Domain Classifier | Target Risk |
| CoDATS | Time Series | Adversarial | Marginal | Domain Classifier, Gradient Reversal Layer | Target Risk |
| AdvSKM | Time Series | Adversarial | Marginal | Spectral Kernel Adversarial MMD | Target Risk |

### 3.3 MODEL SELECTION APPROACHES

Model selection and hyper-parameter tuning for UDA are long standing non-trivial problems due to the absence of target domain labels. However, we find that papers describing 5 out of the 10 included UDA methods use target domain labels to select hyper-parameters, as shown in Table 1, which violates the basic assumption of UDA. In addition, another three papers used fixed hyper-parameters, without describing how these parameters were selected. This issue can lead to unrealistic performance and unfair evaluation. To address this issue, we evaluate multiple realistic model selection approaches that could be applied for TS-UDA without the need of any target domain labels, such as: SRC risk (Ganin et al., 2016) and DEV risk (You et al., 2019). Besides, we design a FST risk, which utilizes affordable few labeled samples from the target domain. Detailed explanation of each risk can be found as follows:

**Source Risk (SRC Risk)**    In this approach, we select the hyper-parameters that achieve the minimum risk on the source domain test set, which is readily available. Hence, this approach can be easily applicable without additional labeling efforts, since it relies on the existing labeled source data (Ganin et al., 2016). However, the effectiveness of the source risk is mainly controlled by the sample size of source data and severity of distribution shift.

**Deep Embedded Validation (DEV Risk)**    This approach tries to yield unbiased estimation to the target risk. Its key idea is to consider the relationship between the source and target data when calculating the risk. Particularly, DEV risk considers the source features that are highly correlated to the target features via importance weighting schemes. By ranking candidate models based on their computed DEV risks, the model with the smallest DEV risk is chosen as the best candidate model to be used for adaptation. However, this approach is computationally expensive and may have unstable performance with smaller source and target datasets.

**Few-Shot Target Risk (FST Risk)**    We propose few-shot target risk as a relaxed target risk. While labeling of vast amount of time series data can be laborious, annotating few-shot samples can still be affordable. We use these samples as a validation set to select the best hyper-parameters. This strategy can achieve similar performance to using target risk, but it does require labeling a few target domain samples.

**Target Risk (TGT Risk)**    This approach involves leaving out a subset of the target domain samples and their labels as a validation set, and using them to select the best performing hyper-parameters on the target domain. Using this model selection technique provides an upper bound on the performance of the UDA method, as it leverages labeled data from the target domain. Even though this risk is impractical in unsupervised settings, it has been used by many UDA papers for model selection (Shu et al., 2018; Saito et al., 2017).

## 4 EXPERIMENTS

### 4.1 BENCHMARKING DATASETS

We evaluate the various UDA algorithms on four benchmark datasets from two real-world applications, namely human activity recognition and sleep stage classification. The benchmark datasets span a range of different characteristics including complexity, type of sensors, samples size, class-distribution, and severity of domain shift, enabling more broad evaluation. The selected datasets are detailed as follows:

**UCIHAR (Anguita et al., 2013)**    UCIHAR is one of the most widely used datasets to evaluate performance on time series data. It contains three different sensors namely, accelerometer, gyroscope, and body sensors. These sensors have been used to collect data from 30 different persons. In our experiments, we treat each subject as a separate domain. Due to the large number of cross-domain combinations, we randomly selected five cross domain scenarios, as in (Liu & Xue, 2021; Wilson et al., 2020).

Table 2: Details of datasets. More details about selected cross-domain scenarios for each dataset cab be found in Tables 4, 5, and 7.

| Dataset | Number of | | | | Total # samples | |
|---------|-----------|--|--|--|-----------------|--|
|         | Users/Domains | Channels | Classes | Timesteps/sample | Training set | Testing set |
| UCIHAR | 32 | 9 | 6 | 128 | 2300 | 990 |
| WISDM | 36 | 3 | 6 | 128 | 1350 | 720 |
| SSC | 20 | 1 | 5 | 3000 | 14280 | 6130 |
| HHAR | 9 | 3 | 6 | 128 | 12716 | 5218 |

**WISDM (Kwapisz et al., 2011)**   WISDM is another popular activity recognition dataset for the evaluation of time series domain adaptation. In this dataset, accelerometer sensors were applied to collect data from 36 subjects. This data can be more challenging because of the class imbalance issue among different subjects. Particularly, some subjects may contain only samples from a subset of the overall classes. Further detailed about ratio of different classes among subjects can be found in the Appendix. Similar to UCIHAR dataset, we consider each subject as a separate domain and we randomly select five cross-domain scenarios.

**SSC (Goldberger et al., 2000)**   Sleep stage classification (SSC) problem aims to classify the electroencephalography (EEG) signals into five stages i.e. Wake (W), Non-Rapid Eye Movement stages (N1, N2, N3), and Rapid Eye Movement (REM). We adopted Sleep-EDF dataset (Goldberger et al., 2000), which contains EEG readings from 20 healthy subjects. We selected a single channel (i.e., Fpz-Cz) following previous studies (Eldele et al., 2021a), and 10 different subjects to construct five subject-wise cross-domain scenarios.

**HHAR (Stisen et al., 2015)**   The Heterogeneity Human Activity Recognition (HHAR) dataset has been collected from 9 different users using sensor readings from smartphones and smartwatches. In our experiments, we consider each user as a domain. We constructed 10 cross-domain scenarios from randomly selected users. We used the same smartphone device for all the selected users to reduce the heterogeneity.

Table 2 summarizes the details of each dataset, e.g., the number of selected domains, the number of sensor channels, the number of classes, the length of each sample, as well as the total number of samples in both training and test portions.

## 4.2   EXPERIMENTAL SETUP

**Hyper-parameters Sweep**   For each algorithm and dataset combination, we conducted extensive random hyper-parameter search with 100 trials. The hyper-parameters are picked by a uniform sampling from a range of predefined values. Details about the specified ranges can be found in Table 8. In addition, for each set of hyper-parameters, we calculated the risk values over three different random seeds, removing the bias to a single seed. We picked the model that achieves the minimum risk value for each model selection strategy. For few-shot target risk experiments, we used five samples per class in each dataset to compute the risk.

**Datasets Preprocessing**   The preprocessing of time series data includes data slicing, train/test splitting, normalization. To promote fair evaluation, we preserved consistency among all the aforementioned processing steps when comparing between different UDA algorithms. We used a sliding window of 128 for human activity recognition datasets. For SSC dataset, we kept the original sample length of 3000 time steps. Next, we split both source and target domains into train/test splits with a ratio of 0.7/0.3. Finally, we normalized both training and testing data based on the training statistics (Wilson et al., 2020; Liu & Xue, 2021). It worth noting that WISDM dataset is severely imbalanced among different subjects (see Figure 4(b) in the appendix), with some classes not being available for some subjects. To stabilize the training, we selected subjects that contain samples from all the classes when constructing the source and target domains.

**Backbone Network** To ensure a fair evaluation among all the baselines, we used a fixed backbone network among all the compared UDA methods. We employed a 3-layer 1D-CNN with an adaptive average pooling, to handle signals with different lengths, as a feature extractor. This architecture is commonly used in series applications (Wilson et al., 2020; Eldele et al., 2021b). We adopted a single layer fully connected network as the classifier. Yet, to include CoDATS in our experiments, we used a 3-layer fully connected classifier (as proposed in their work) to differentiate it from DANN.

**Training Procedure** All the training procedures have been standardized across all UDA algorithms. For instance, we trained each models for 40 epochs, as performance tends to decrease with longer training. We reported the model performance after completing the last epoch. Regarding the optimization of the model, we used Adam optimizer with a fixed weight decay of 1e-4 and $(\beta_1, \beta_2) = (0.5, 0.99)$. The learning rate was set to be a tunable hyper-parameter for each method on each dataset. We excluded any learning rate scheduling schemes from our experiments to ensure that the contribution is mainly attributed to the selected UDA algorithm.

## 5 RESULTS DISCUSSION

We conducted extensive experiments for each UDA method on all the datasets. The average F1-score of the five cross-domain scenarios for each dataset-algorithm combination is reported in Table 3. We also show the results of each risk to compare the performance to the target risk. Detailed versions of Table 3 can be found in Tables 4, 5, and 7 in the appendix. We further discuss the main findings in the following paragraphs.

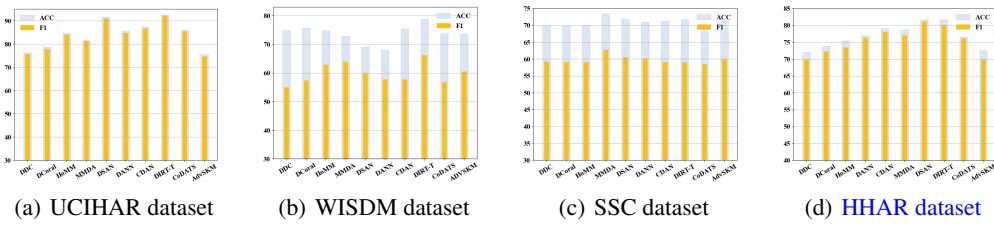

(a) UCIHAR dataset  (b) WISDM dataset  (c) SSC dataset  (d) HHAR dataset

Figure 2: Results of best models according to target risk for different methods in terms of accuracy and macro average F1-score.

**Visual UDA methods achieve comparable performance to TS-UDA methods on time series data.** Table 3 shows the performance of the adapted visual UDA methods along with the existing TS-UDA methods. Surprisingly, with standardizing the backbone network, we find that visual UDA methods achieve competitive or even better performance than TS-UDA methods. This finding is consistent for all the model selection strategies across the benchmarking datasets. A possible explanation is that all selected UDA algorithms are applied on the vectorized output features of the backbone network, which can be independent from the input data modality. This finding suggests that with a standard backbone network, visual UDA algorithms can be strong baselines for TS-UDA.

**Methods with joint distribution alignment tend to perform consistently better.** As shown in Table 1, methods that address the joint distributions (i.e., both marginal and conditional distributions concurrently) such as DIRT-T, MMDA, and DSAN, achieve the state-of-the-art performance on the four datasets, outperforming TS-UDA methods. For example, the best performing method, as selected by the target risk, is DIRT-T in both UCIHAR and WISDM datasets, and DSAN in SSC and HHAR datasets. Similarly, with respect to the different risks, DIRT-T, MMDA, and DSAN are interchangeably achieving performing best across the benchmarking datasets. Hence, considering the conditional distribution when aligning the source and target can be beneficial to the performance.

**Model selection has a significant effect on the performance.** Model selection strategies can yield different performance for the same UDA algorithm, as shown in Table 3. This reveals the significant contribution of the model selection approach to the overall performance. While target risk achieves superior performance, it is infeasible for practical scenarios with the absence of target

Table 3: The average results (from 5 cross-domain scenarios) according to the minimum risk value in terms of MF1-score.

| Dataset | Algorithm | SRC Risk | DEV Risk | FST Risk | TGT Risk |
|---------|-----------|----------|----------|----------|----------|
| UCIHAR | DDC | 68.83 | 74.14 | 74.25 | 75.67 |
|  | Deep-Coral | 71.99 | 71.43 | 77.23 | 77.71 |
|  | HoMM | 75.86 | 78.28 | 81.3 | 84.10 |
|  | MMDA | 80.12 | 80.12 | 79.54 | 81.40 |
|  | DSAN | 83.31 | 81.07 | _87.13_ | _90.96_ |
|  | DANN | 79.82 | _80.89_ | 83.1 | 84.97 |
|  | CDAN | _86.55_ | 64.66 | 86.79 | 86.79 |
|  | DIRT-T | **86.72** | **82.54** | **88.47** | **92.20** |
|  | CoDATS | 79.05 | 65.12 | 79.1 | 85.47 |
|  | AdvSKM | 71.08 | 74.62 | 74.47 | 74.67 |
|  | Avg/risk | 78.33 | 75.29 | 81.14 | 83.39 |
| WISDM | DDC | 54.98 | 52.80 | 50.05 | 55.03 |
|  | Deep-Coral | 55.54 | 53.85 | 49.45 | 57.43 |
|  | HoMM | 57.49 | **61.23** | 46.56 | 62.98 |
|  | MMDA | **57.53** | 57.30 | 52.12 | _63.97_ |
|  | DSAN | 56.51 | 56.51 | _53.41_ | 60.08 |
|  | DANN | 53.21 | 54.48 | 49.45 | 57.81 |
|  | CDAN | 52.49 | 53.27 | 52.75 | 57.85 |
|  | DIRT-T | 60.43 | 53.24 | **62.61** | **66.28** |
|  | CoDATS | 52.72 | _54.27_ | 48.64 | 56.57 |
|  | AdvSKM | _53.95_ | 57.46 | 49.02 | 60.55 |
|  | Avg/risk | 53.93 | 53.52 | 51.91 | 60.82 |
| SSC | DDC | 59.18 | 59.21 | **59.22** | 59.22 |
|  | Deep-Coral | 59.12 | 58.81 | _58.82_ | 59.12 |
|  | HoMM | 59.06 | 60.95 | 58.70 | 59.06 |
|  | MMDA | **62.08** | _61.49_ | 57.98 | _62.79_ |
|  | DSAN | _58.14_ | **59.85** | 58.97 | **60.57** |
|  | DANN | 60.26 | 57.77 | 60.26 | 60.26 |
|  | CDAN | 54.89 | 56.86 | 56.17 | 59.04 |
|  | DIRT-T | 58.44 | 59.26 | 58.23 | 59.42 |
|  | CoDATS | 56.76 | 55.79 | 54.64 | 58.44 |
|  | AdvSKM | 59.94 | 59.92 | 59.93 | 60.21 |
|  | Avg/risk | 59.37 | 59.34 | 59.10 | 60.30 |
| HHAR | Deep-Coral | 70.78 | 69.88 | 70.68 | 72.28 |
|  | HoMM | 71.18 | 72.50 | 68.62 | 73.47 |
|  | MMDA | 66.20 | 70.23 | 71.07 | 77.04 |
|  | DSAN | 76.18 | **78.95** | _78.18_ | **81.14** |
|  | DANN | _76.24_ | 72.62 | 73.68 | 76.42 |
|  | CDAN | **77.74** | 77.43 | 77.43 | 78.09 |
|  | DIRT-T | 75.56 | _78.69_ | **78.41** | _80.04_ |
|  | CoDATS | 75.11 | _73.72_ | 74.74 | _76.09_ |
|  | ADVSKM | 66.58 | 66.96 | 69.93 | 69.93 |
|  | Avg/risk | 72.32 | 72.73 | 73.20 | 75.44 |

domain labels. Surprisingly, source risk can achieve a comparable performance to the target risk on UCIHAR, HHAR and SSC datasets. In addition, with affordable labeling efforts, our proposed few-shot target risk can also achieve a competitive performance to the target risk. However, for severely imbalanced WISDM dataset, different model selection approaches fail to follow the target risk performance.

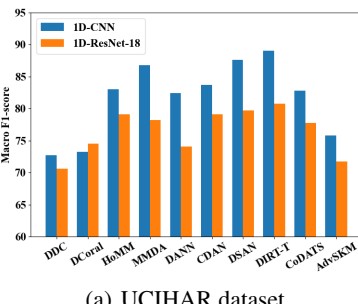

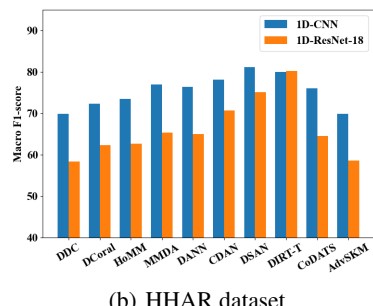

(a) UCIHAR dataset             (b) HHAR dataset

Figure 3: Comparison between 1D-CNN and 1D-ResNet-18 backbones applied on the both UCI-HAR and HHAR dataset. Results are in terms of macro F1-score.

**Backbone network can be pivotal to the performance.** To investigate the impact of the backbone network on the method's performance, we evaluate all the UDA algorithms under two different backbone networks for both small-scale and large-scale datasets, i.e., UCIHAR and HHAR. Particularly, we employ 1D-CNN (described in Section 4.2) and 1D-ResNet-18 (i.e., a standard ResNet-18 with 1D convolutional kernel) as backbone networks. The two architectures are different in terms of their complexity and the number of trainable parameters. Figure 3 shows a comparison of different UDA algorithms under two backbone architectures. Clearly, the two backbone networks can have different absolute performance for each UDA methods. Nevertheless, the relative performance between different UDA methods can still be consistent when fixing the backbone network. For instance in Figure 3b, the DSAN and DIRT approaches perform best for both 1D-CNN and 1D-ResNet-18 backbone networks. Besides, it is still clear that visual UDA methods perform better than TS-UDA methods with the two different backbone networks.

**Accuracy metric should not be used to measure performance for imbalanced data.** While it is well-known that accuracy is not a representative metric for class-imbalanced datasets, existing TS-UDA methods are still using it to report their performance (Chang et al., 2020; Wilson et al., 2020; Liu & Xue, 2021). We aim to re-emphasize that the accuracy metric can give over-optimistic results when considering the imbalanced nature in most time series data. For example, in Figure 2(b), although CDAN achieves higher accuracy than some other methods such as DDC, MMDA and DSAN, it performs the worst in terms of F1-score. In contrast, with a balanced dataset (i.e., UCI-HAR), accuracy can still be representative and achieve similar performance to F1-Score, as shown in Figure 3(a).

## 6  CONCLUSIONS AND RECOMMENDATIONS

In this work, we provided ADATIME, a systematic evaluation methodology for evaluating the existing domain adaptation methods on time series data. To ensure fair and realistic evaluation, we standardized the benchmarking dataset, evaluation schemes, and backbone networks among different domain adaptation methods. Moreover, we explored more realistic model selection approaches that can work with without any target domain labels or only few-shot labeled samples. Based on our systematic study and unveiled findings, we suggest the following recommendations:

- Future research on time series domain adaptation should consider visual UDA methods as strong baselines.
- Considerable amount of target labels should not be used for the UDA model selection. Instead, source risk or few-shot target risk can be better candidates for a more realistic model selection approach.
- The backbone network should be fixed when comparing between different UDA methods.
- Larger datasets should be considered when comparing baselines to obtain reliable results.
- Accuracy metric should no longer be used to evaluate imbalance datasets, as it can yield misleading results.

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

# A DETAILED RESULTS OF OUR FRAMEWORK

Here, we provide the detailed results including mean and standard deviation for each cross-domain scenario. It can be clearly seen, that datasets with small sample size suffers from high variance among the different cross-domain scenarios.

Table 4: Detailed results of scenarios of UCIHAR dataset in terms of MF1 score.

| Risk | Algorithm | 0→11 | 12→16 | 9→18 | 6→23 | 7→13 | AVG |
|------|-----------|------|-------|------|------|------|-----|
| Target | DDC | 60.00±13.32 | 66.77±8.46 | 61.41±5.80 | 88.55±1.42 | 77.29±2.11 | 75.67 |
| | Deep-Coral | 67.2±13.67 | 64.58±8.72 | 54.38±9.69 | 89.66±2.54 | 90.46±2.96 | 77.71 |
| | HoMM | 83.54±2.99 | 63.45±2.07 | 71.25±4.42 | 94.97±2.49 | 91.41±1.33 | 84.10 |
| | MMDA | 72.91±2.78 | 74.64±2.88 | 62.62±2.63 | 91.14±0.46 | 90.61±2.00 | 81.40 |
| | DSAN | 99.23±1.09 | 65.71±2.92 | 89.69±7.44 | 97.82±1.54 | 94.33±0.00 | 90.96 |
| | DANN | 98.09±1.68 | 62.08±1.69 | 70.7±11.36 | 85.6±15.71 | 93.33±0.00 | 84.97 |
| | CDAN | 98.19±1.57 | 61.20±3.27 | 71.3±14.64 | 96.73±0.00 | 93.33±0.00 | 86.79 |
| | DIRT-T | 98.13±2.64 | 82.05±8.61 | 85.90±6.63 | 93.76±3.10 | 93.35±0.00 | 92.20 |
| | CoDATS | 86.65±4.28 | 61.03±2.33 | 80.51±8.47 | 92.08±4.39 | 92.61±0.51 | 85.47 |
| | AdvSKM | 65.74±2.69 | 60.52±1.99 | 53.25±5.19 | 79.63±8.52 | 88.89±3.12 | 74.67 |
| Few-shot | DDC | 67.95±7.51 | 58.58±1.55 | 48.33±4.06 | 84.34±5.22 | 86.33±2.19 | 74.25 |
| | Deep-Coral | 70.86±5.85 | 59.3±0.77 | 58.5±10.7 | 89.5±1.6 | 85.23±3.01 | 77.23 |
| | HoMM | 78.87±8.37 | 60.34±1.07 | 66.97±3.41 | 93.84±2.04 | 87.82±4.43 | 81.3 |
| | MMDA | 74.36±9.26 | 66.01±5.15 | 54.92±4.27 | 95.88±1.2 | 93.33±0.0 | 79.54 |
| | DSAN | 89.47±8.76 | 65.97±2.59 | 78.02±7.52 | 96.68±2.74 | 92.61±0.51 | 87.13 |
| | DANN | 87.73±5.29 | 60.33±1.9 | 69.69±9.83 | 89.88±6.34 | 93.33±0.0 | 83.10 |
| | CDAN | 98.19±1.57 | 61.2±3.27 | 71.31±14.64 | 96.73±0.0 | 93.33±0.0 | 86.79 |
| | DIRT-T | 92.33±1.49 | 71.63±6.64 | 86.87±0.39 | 86.71±14.17 | 93.33±0.0 | 88.47 |
| | CoDATS | 71.6±15.34 | 65.1±0.68 | 64.51±14.12 | 92.04±4.05 | 81.41±6.04 | 79.10 |
| | AdvSKM | 64.45±2.59 | 61.99±4.07 | 53.13±4.59 | 78.09±9.92 | 88.89±3.12 | 74.47 |
| DEV | DDC | 72.0±3.51 | 59.65±4.11 | 45.42±5.89 | 86.14±1.97 | 81.66±8.07 | 74.14 |
| | Deep-Coral | 67.55±11.65 | 62.13±7.74 | 47.77±2.84 | 72.44±13.45 | 78.7±18.24 | 71.43 |
| | HoMM | 73.38±7.34 | 59.84±1.43 | 60.02±11.83 | 90.48±0.8 | 85.94±2.52 | 78.28 |
| | MMDA | 83.22±3.46 | 62.64±10.42 | 58.43±2.55 | 96.73±0.0 | 94.12±1.11 | 80.12 |
| | DSAN | 75.58±9.18 | 61.71±1.75 | 67.1±4.61 | 93.22±2.49 | 88.82±3.08 | 81.07 |
| | DANN | 77.77±18.26 | 63.26±2.49 | 57.49±7.77 | 95.86±1.84 | 91.71±0.84 | 80.89 |
| | CDAN | 71.51±8.84 | 54.66±2.91 | 40.94±3.18 | 61.31±9.02 | 82.06±11.91 | 64.66 |
| | DIRT-T | 88.44±9.23 | 58.47±2.98 | 65.89±13.25 | 90.56±8.73 | 93.73±0.56 | 82.54 |
| | CoDATS | 51.81±4.67 | 54.81±2.76 | 31.83±8.89 | 81.23±4.07 | 80.98±13.74 | 65.12 |
| | AdvSKM | 65.74±2.69 | 60.09±1.4 | 53.7±4.61 | 79.31±8.95 | 88.89±3.12 | 74.62 |
| Source | DDC | 53.28±5.44 | 64.59±6.34 | 41.99±1.47 | 89.01±2.14 | 85.65±7.92 | 68.83 |
| | Deep-Coral | 62.42±1.97 | 62.19±4.68 | 31.4±7.7 | 88.42±1.01 | 87.84±2.78 | 71.99 |
| | HoMM | 62.95±20.6 | 59.82±0.4 | 53.14±4.71 | 90.04±4.45 | 89.59±4.01 | 75.86 |
| | MMDA | 83.22±3.46 | 62.64±10.42 | 58.43±2.55 | 96.73±0.0 | 94.12±1.11 | 80.12 |
| | DSAN | 84.81±13.2 | 62.91±0.39 | 77.63±0.94 | 81.91±23.31 | 92.97±0.51 | 83.31 |
| | DANN | 70.28±2.86 | 65.45±4.76 | 71.34±6.17 | 90.0±2.71 | 90.51±0.86 | 79.82 |
| | CDAN | 88.73±4.7 | 60.25±4.61 | 81.39±6.15 | 96.73±0.0 | 92.97±0.51 | 86.55 |
| | DIRT-T | 88.44±9.23 | 61.36±2.41 | 78.88±4.12 | 98.64±1.39 | 93.01±0.46 | 86.72 |
| | CoDATS | 71.45±12.93 | 61.98±2.88 | 72.11±4.98 | 96.12±0.86 | 78.92±12.71 | 79.05 |
| | AdvSKM | 69.97±3.68 | 53.73±11.04 | 35.46±8.02 | 85.73±6.11 | 82.29±4.67 | 71.08 |

Table 5: Detailed results of scenarios of WISDM dataset in terms of MF1 score.

| Risk | Algorithm | 35→31 | 7→18 | 20→30 | 6→19 | 18→23 | AVG |
|------|-----------|-------|------|-------|------|-------|-----|
| Target | DDC | 51.84±5.68 | 43.56±0.66 | 65.83±2.04 | 62.58±2.72 | 51.36±10.81 | 55.03 |
| | Deep Coral | 56.48±8.18 | 44.32±1.79 | 66.33±2.17 | 63.11±3.17 | 56.91±3.24 | 57.43 |
| | HoMM | 58.49±7.79 | 53.64±3.48 | 70.94±2.8 | 74.88±6.55 | 56.96±24.07 | 62.98 |
| | MMDA | 54.37±5.23 | 47.75±2.04 | 66.35±2.1 | 76.68±5.71 | 74.7±2.28 | 63.97 |
| | DSAN | 70.35±1.44 | 44.77±4.4 | 69.53±4.91 | 49.85±7.48 | 65.88±7.21 | 60.08 |
| | DANN | 62.53±6.97 | 43.55±2.06 | 53.96±3.03 | 79.05±15.62 | 49.96±5.83 | 57.81 |
| | CDAN | 60.93±8.6 | 58.98±5.36 | 60.43±11.19 | 59.61±0.89 | 49.3±1.05 | 57.85 |
| | DIRT-T | 68.59±10.7 | 73.57±9.09 | 72.06±8.6 | 50.14±7.43 | 67.03±2.41 | 66.28 |
| | CODATS | 50.93±5.46 | 51.21±5.66 | 67.28±3.2 | 65.24±3.68 | 49.1±10.92 | 56.75 |
| | AdvSKM | 55.15±9.56 | 52.36±2.73 | 72.83±3.69 | 58.64±4.33 | 63.78±10.2 | 60.55 |
| Few-shot | DDC | 38.86±14.69 | 43.3±1.8 | 56.56±0.93 | 62.43±7.42 | 49.11±0.23 | 50.05 |
| | Deep Coral | 39.37±15.11 | 45.62±4.25 | 56.86±0.72 | 56.46±5.43 | 48.95±0.19 | 49.45 |
| | HoMM | 35.41±13.8 | 30.52±3.6 | 60.99±1.1 | 54.74±7.85 | 51.15±7.39 | 46.56 |
| | MMDA | 43.47±4.95 | 57.9±1.84 | 55.9±2.97 | 59.8±8.45 | 43.52±8.41 | 52.12 |
| | DSAN | 52.92±14.31 | 51.98±4.93 | 64.35±3.78 | 49.26±7.09 | 48.51±8.15 | 53.41 |
| | DANN | 43.11±9.84 | 43.08±1.21 | 57.63±2.53 | 55.2±4.93 | 48.23±0.4 | 49.45 |
| | CDAN | 54.53±1.31 | 57.03±0.53 | 64.55±6.06 | 39.76±7.63 | 47.85±0.46 | 52.75 |
| | DIRT-T | 62.52±10.06 | 68.8±7.94 | 62.31±5.49 | 51.86±6.68 | 67.56±0.51 | 62.61 |
| | CODATS | 40.68±24.05 | 37.82±2.61 | 61.01±1.2 | 56.3±8.87 | 47.38±1.23 | 48.64 |
| | AdvSKM | 57.43±12.5 | 73.58±1.33 | 71.2±3.2 | 78.28±3.05 | 67.78±0.79 | 49.02 |
| DEV | DDC | 48.85±16.15 | 45.17±6.65 | 70.04±10.28 | 57.51±6.89 | 42.4±8.53 | 52.80 |
| | DeepCoral | 42.36±11.34 | 47.07±7.25 | 67.16±4.86 | 65.06±3.76 | 47.57±8.73 | 53.85 |
| | HoMM | 66.29±0.84 | 48.67±6.31 | 65.3±2.45 | 63.78±4.35 | 62.11±7.57 | 61.23 |
| | MMDA | 60.34±7.52 | 41.58±8.79 | 64.39±4.28 | 55.74±3.88 | 64.47±10.75 | 57.30 |
| | DSAN | 57.25±6.07 | 52.77±2.23 | 63.4±0.7 | 53.35±5.37 | 55.76±1.46 | 56.51 |
| | DANN | 52.21±1.09 | 41.16±6.62 | 71.96±10.1 | 59.09±3.57 | 48.0±0.9 | 54.48 |
| | CDAN | 49.02±4.2 | 57.65±0.18 | 65.5±0.61 | 44.03±0.81 | 50.16±0.44 | 53.27 |
| | DIRT-T | 46.75±3.54 | 57.89±0.15 | 65.49±0.62 | 45.16±0.0 | 50.9±0.4 | 53.24 |
| | CODATS | 40.96±19.0 | 42.0±3.75 | 69.65±7.6 | 70.59±12.51 | 48.15±15.11 | 54.27 |
| | AdvSKM | 61.91±6.95 | 49.84±5.31 | 69.35±1.38 | 54.89±4.14 | 51.3±10.33 | 57.46 |
| Source | DDC | 51.47±5.69 | 43.65±0.78 | 65.83±2.04 | 62.58±2.72 | 51.36±10.81 | 54.98 |
| | Deep Coral | 53.46±7.12 | 43.65±0.78 | 66.08±2.05 | 63.16±3.14 | 51.36±10.81 | 55.54 |
| | HoMM | 57.94±7.0 | 43.23±0.53 | 65.47±1.13 | 63.91±4.12 | 56.91±3.24 | 57.49 |
| | MMDA | 61.44±6.08 | 49.79±6.76 | 67.82±1.37 | 60.83±0.27 | 47.78±5.98 | 57.53 |
| | DSAN | 57.25±6.07 | 52.77±2.23 | 63.4±0.7 | 53.35±5.37 | 55.76±1.46 | 56.51 |
| | DANN | 46.56±11.92 | 44.04±2.61 | 68.13±1.27 | 54.41±4.61 | 52.92±2.86 | 53.21 |
| | CDAN | 44.25±6.84 | 57.44±7.58 | 63.67±0.63 | 47.81±5.88 | 49.27±0.12 | 52.49 |
| | DIRT-T | 72.59±4.03 | 57.74±0.06 | 53.47±2.17 | 60.91±0.38 | 57.46±8.41 | 60.43 |
| | CODATS | 74.06±4.77 | 35.8±0.78 | 54.21±4.5 | 45.54±1.33 | 54.0±13.2 | 52.72 |
| | AdvSKM | 42.3±16.63 | 53.66±5.72 | 62.59±2.7 | 60.37±0.81 | 50.84±2.66 | 53.95 |

Table 6: Detailed results of scenarios of SSC dataset in terms of MF1 score.

| Risk | Algorithm | 16→1 | 9→14 | 12→5 | 7→18 | 0→11 | AVG |
|------|-----------|------|------|------|------|------|-----|
| Target | DDC | 55.47±1.72 | 63.57±1.43 | 55.43±2.75 | 67.46±1.45 | 54.17±1.79 | 59.22 |
| | Deep-Coral | 55.50±1.74 | 63.50±1.36 | 55.35±2.64 | 67.49±1.50 | 53.76±1.89 | 59.12 |
| | HoMM | 55.51±1.79 | 63.49±1.14 | 55.46±2.71 | 67.50±1.50 | 53.37±2.47 | 59.06 |
| | MMDA | 62.92±0.96 | 71.04±2.39 | 65.84±1.08 | 70.95±0.82 | 43.23±4.31 | 62.79 |
| | DSAN | 59.87±2.84 | 70.71±2.79 | 65.55±0.79 | 68.44±1.39 | 38.28±3.57 | 60.57 |
| | DANN | 58.68±3.29 | 64.29±1.08 | 64.65±1.83 | 69.54±3.00 | 44.13±5.84 | 60.26 |
| | CDAN | 59.65±4.96 | 64.18±6.37 | 64.43±1.17 | 67.61±3.55 | 39.38±3.28 | 59.04 |
| | DIRT-T | 61.31±4.23 | 66.39±4.86 | 66.95±1.72 | 70.51±0.89 | 33.05±2.49 | 59.42 |
| | CoDATS | 63.84±3.36 | 63.51±6.92 | 52.54±5.94 | 66.06±2.48 | 46.28±5.99 | 58.44 |
| | AdvSKM | 57.83±1.42 | 64.76±3.0 | 55.73±1.42 | 67.58±3.64 | 55.2±4.19 | 60.21 |
| Few-shot | DDC | 55.48±1.76 | 63.54±1.33 | 55.32±2.94 | 67.5±1.5 | 54.28±1.68 | 59.22 |
| | Deep-Coral | 55.5±1.84 | 63.55±1.33 | 55.42±2.66 | 67.53±1.54 | 52.1±2.85 | 58.82 |
| | HoMM | 55.51±1.79 | 63.5±1.14 | 55.46±2.71 | 67.5±1.5 | 53.37±2.47 | 59.06 |
| | MMDA | 65.63±0.67 | 65.92±4.44 | 57.99±6.43 | 71.5±0.97 | 28.9±3.78 | 57.98 |
| | DSAN | 56.39±0.67 | 63.85±0.63 | 62.47±2.6 | 68.92±1.67 | 43.25±2.76 | 58.97 |
| | DANN | 58.68±3.3 | 64.3±1.08 | 64.65±1.83 | 69.54±3.0 | 44.13±5.84 | 60.26 |
| | CDAN | 59.87±2.67 | 63.55±3.16 | 62.13±1.8 | 64.12±0.48 | 31.19±8.26 | 56.17 |
| | DIRT-T | 56.33±5.86 | 65.15±1.99 | 64.88±5.58 | 69.83±1.57 | 34.99±0.54 | 58.23 |
| | CoDATS | 59.84±0.64 | 53.02±4.53 | 57.58±1.75 | 55.12±3.55 | 47.64±2.4 | 54.64 |
| | AdvSKM | 57.68±0.79 | 64.31±2.93 | 55.29±2.58 | 67.22±3.9 | 55.16±4.39 | 59.93 |
| DEV | DDC | 55.53±1.87 | 63.57±1.26 | 55.35±2.73 | 67.46±1.55 | 54.14±1.7 | 59.21 |
| | Deep-Coral | 55.5±1.84 | 63.55±1.33 | 55.42±2.66 | 67.5±1.5 | 52.1±2.85 | 58.81 |
| | HoMM | 55.57±2.0 | 63.66±1.48 | 55.87±2.93 | 67.49±1.51 | 50.93±4.31 | 58.70 |
| | MMDA | 63.44±1.49 | 67.14±4.78 | 64.93±1.21 | 71.89±1.44 | 39.88±4.96 | 61.49 |
| | DSAN | 58.76±2.02 | 69.45±4.04 | 64.92±1.65 | 68.69±0.99 | 37.43±2.9 | 59.85 |
| | DANN | 58.78±4.76 | 64.61±0.93 | 65.47±0.95 | 68.88±2.81 | 31.13±1.74 | 57.77 |
| | CDAN | 60.95±1.13 | 60.54±10.01 | 65.0±1.34 | 67.02±1.13 | 30.79±10.69 | 56.86 |
| | DIRT-T | 54.42±12.46 | 71.33±3.72 | 64.99±4.98 | 69.94±0.43 | 35.62±3.79 | 59.26 |
| | CoDATS | 60.03±1.18 | 52.22±10.55 | 56.96±2.4 | 68.64±2.93 | 41.1±5.14 | 55.79 |
| | AdvSKM | 57.8±0.69 | 64.27±2.93 | 55.12±2.52 | 67.31±3.83 | 55.11±4.56 | 59.92 |
| Source | DDC | 55.48±1.76 | 63.57±1.29 | 55.16±2.76 | 67.5±1.5 | 54.24±1.79 | 59.18 |
| | Deep-Coral | 55.5±1.74 | 63.5±1.36 | 55.35±2.64 | 67.5±1.5 | 53.76±1.89 | 59.12 |
| | HoMM | 55.51±1.79 | 63.5±1.14 | 55.46±2.71 | 67.5±1.5 | 53.37±2.47 | 59.06 |
| | MMDA | 59.6±0.51 | 68.25±4.17 | 65.63±0.85 | 71.06±0.99 | 45.89±1.97 | 62.08 |
| | DSAN | 63.05±3.14 | 63.84±10.11 | 57.55±11.16 | 68.84±2.25 | 37.46±4.76 | 58.14 |
| | DANN | 58.68±3.3 | 64.3±1.08 | 64.65±1.83 | 69.54±3.0 | 44.13±5.84 | 60.26 |
| | CDAN | 62.06±0.91 | 63.32±5.02 | 48.8±1.02 | 63.46±1.18 | 36.86±8.23 | 54.89 |
| | DIRT-T | 59.11±3.24 | 65.08±1.42 | 65.5±4.92 | 67.27±1.58 | 35.29±2.92 | 58.44 |
| | CoDATS | 56.52±1.76 | 68.2±5.72 | 59.72±6.66 | 63.31±3.9 | 36.05±8.95 | 56.76 |
| | AdvSKM | 57.78±0.72 | 64.29±2.97 | 55.15±2.52 | 67.33±3.82 | 55.16±4.4 | 59.94 |

Table 7: Detailed results of scenarios of HHAR dataset in terms of MF1 score.

| Risk | Algorithm | 0→6 | 1→6 | 2→7 | 3→8 | 4→5 | AVG |
|------|-----------|-----|-----|-----|-----|-----|-----|
| Target | DDC | 51.22±14.18 | 85.11±7.11 | 48.6±5.66 | 77.43±2.47 | 86.97±1.85 | 69.87±0 |
| | Deep-Coral | 57.64±4.59 | 89.81±0.33 | 44.15±0.92 | 79.3±0.3 | 90.53±3.05 | 72.28±0 |
| | HoMM | 64.85±0.96 | 89.12±0.61 | 44.44±0.6 | 80.2±1.1 | 88.73±3.01 | 73.47±0 |
| | MMDA | 61.66±4.66 | 90.85±0.51 | 53.36±8.87 | 88.07±5.29 | 91.24±4.96 | 77.04±0 |
| | DSAN | 56.47±10.89 | 92.77±1.03 | 61.07±1.46 | 98.14±0.51 | 97.26±0.47 | 81.14±0 |
| | DANN | 47.02±0.57 | 93.02±1.9 | 49.06±8.38 | 95.77±1.91 | 97.24±0.51 | 76.42±0 |
| | CDAN | 56.52±8.35 | 92.4±0.76 | 50.76±6.24 | 93.09±9.94 | 97.67±0.48 | 78.09±0 |
| | DIRT-T | 64.5±9.4 | 94.84±1.52 | 59.9±13.38 | 83.26±2.25 | 97.73±0.47 | 80.04±0 |
| | CoDATS | 46.45±0.64 | 92.59±0.71 | 48.13±8.96 | 96.89±1.97 | 96.38±2.42 | 76.09±0 |
| | ADVSKM | 59.39±4.59 | 81.43±5.52 | 47.75±3.98 | 79.05±0.42 | 82.03±3.24 | 69.93±0 |
| Few-shot | DDC | 61.61±1.95 | 78.86±14.46 | 47.77±4.89 | 78.4±1.31 | 79.65±3.13 | 69.26±0 |
| | Deep-Coral | 59.35±4.8 | 86.58±6.14 | 44.8±2.79 | 77.65±2.23 | 85.04±6.2 | 70.68±0 |
| | HoMM | 54.97±5.29 | 84.99±9.19 | 41.65±1.86 | 78.38±1.9 | 83.12±9.49 | 68.62±0 |
| | MMDA | 60.13±6.66 | 84.15±10.34 | 55.47±4.41 | 80.31±10.76 | 75.26±4.33 | 71.07±0 |
| | DSAN | 52.92±16.13 | 92.67±1.39 | 50.85±10.21 | 97.11±0.39 | 97.33±0.84 | 78.18±0 |
| | DANN | 53.84±6.34 | 86.38±12.08 | 57.48±1.61 | 78.94±6.82 | 91.78±8.26 | 73.68±0 |
| | CDAN | 45.52±0.9 | 92.99±0.7 | 54.1±7.12 | 98.17±0.37 | 96.39±1.37 | 77.43±0 |
| | DIRT-T | 54.88±15.6 | 94.05±1.3 | 64.63±0.3 | 80.6±0.45 | 97.9±0.68 | 78.41±0 |
| | CoDATS | 44.72±5.1 | 93.61±0.7 | 53.33±7.71 | 93.52±1.67 | 88.51±6.23 | 74.74±0 |
| | ADVSKM | 56.25±7.15 | 82.68±3.1 | 45.91±5.88 | 76.62±5.49 | 83.84±2.96 | 69.06±0 |
| DEV | DDC | 62.61±1.32 | 73.99±9.45 | 43.61±0.89 | 76.24±2.53 | 75.17±5.66 | 66.32±0 |
| | Deep-Coral | 54.82±9.16 | 89.31±1.44 | 48.44±1.98 | 77.39±2.81 | 79.44±4.64 | 69.88±0 |
| | HoMM | 63.58±2.24 | 88.49±2 | 47.12±4.27 | 79.23±1.13 | 84.07±1.19 | 72.5±0 |
| | MMDA | 59.52±3.77 | 86.53±2.06 | 48.99±10.42 | 77.8±2.28 | 78.3±7.36 | 70.22±0 |
| | DSAN | 58.81±7.19 | 93.42±0.64 | 45.61±0.5 | 98.44±0.23 | 98.47±0.32 | 78.95±0 |
| | DANN | 46.54±0.61 | 90.73±1.97 | 46.58±3.13 | 83.43±10.12 | 95.83±0.28 | 72.62±0 |
| | CDAN | 45.52±0.9 | 92.99±0.7 | 54.1±7.12 | 98.17±0.37 | 96.39±1.37 | 77.43±0 |
| | DIRT-T | 52.63±9.77 | 93.1±2.06 | 63.49±1.95 | 87.08±10.06 | 97.13±0.44 | 78.69±0 |
| | CoDATS | 44.7±1.65 | 91.98±1.01 | 47.56±5.04 | 91.83±4.56 | 92.52±3.14 | 73.72±0 |
| | ADVSKM | 45.52±0.9 | 92.99±0.7 | 54.1±7.12 | 98.17±0.37 | 96.39±1.37 | 77.43±0 |
| Source | DDC | 62.18±1.56 | 79.2±9.5 | 44.53±1.32 | 76.65±1.83 | 75.53±4.9 | 67.62±0 |
| | Deep-Coral | 63.14±1.57 | 88.27±3.02 | 44.59±0.43 | 78.33±1.77 | 79.55±2.64 | 70.78±0 |
| | HoMM | 63.14±2.11 | 87.58±2.59 | 47.27±4.92 | 77.62±1.67 | 80.28±0.56 | 71.18±0 |
| | MMDA | 65.42±1.48 | 69.07±1.49 | 41.67±0.96 | 76.62±3.05 | 78.2±6.97 | 66.2±0 |
| | DSAN | 56.42±8.91 | 93±0.54 | 49.68±7.7 | 84.28±12.64 | 97.53±0.3 | 76.18±0 |
| | DANN | 61.04±3.52 | 91.78±0.61 | 53.44±2.85 | 80.95±1.68 | 94±1.88 | 76.24±0 |
| | CDAN | 45.52±0.9 | 92.99±0.7 | 54.1±7.12 | 98.17±0.37 | 96.39±1.37 | 77.43±0 |
| | DIRT-T | 47.26±0.16 | 94.06±0.81 | 57.55±9.64 | 81.14±0.15 | 97.78±0.78 | 75.56±0 |
| | CoDATS | 52.32±9.64 | 93.36±0.11 | 43.67±0.86 | 91.06±11.21 | 95.16±2.16 | 75.11±0 |
| | ADVSKM | 56.08±2.45 | 76.28±8.67 | 38.54±9.01 | 79.73±1.33 | 82.29±5.25 | 66.58±0 |

# B CLASS DISTRIBUTION OF DIFFERENT SUBJECTS

In this section, we visualize the class distribution of each selected subjects for all the datasets.

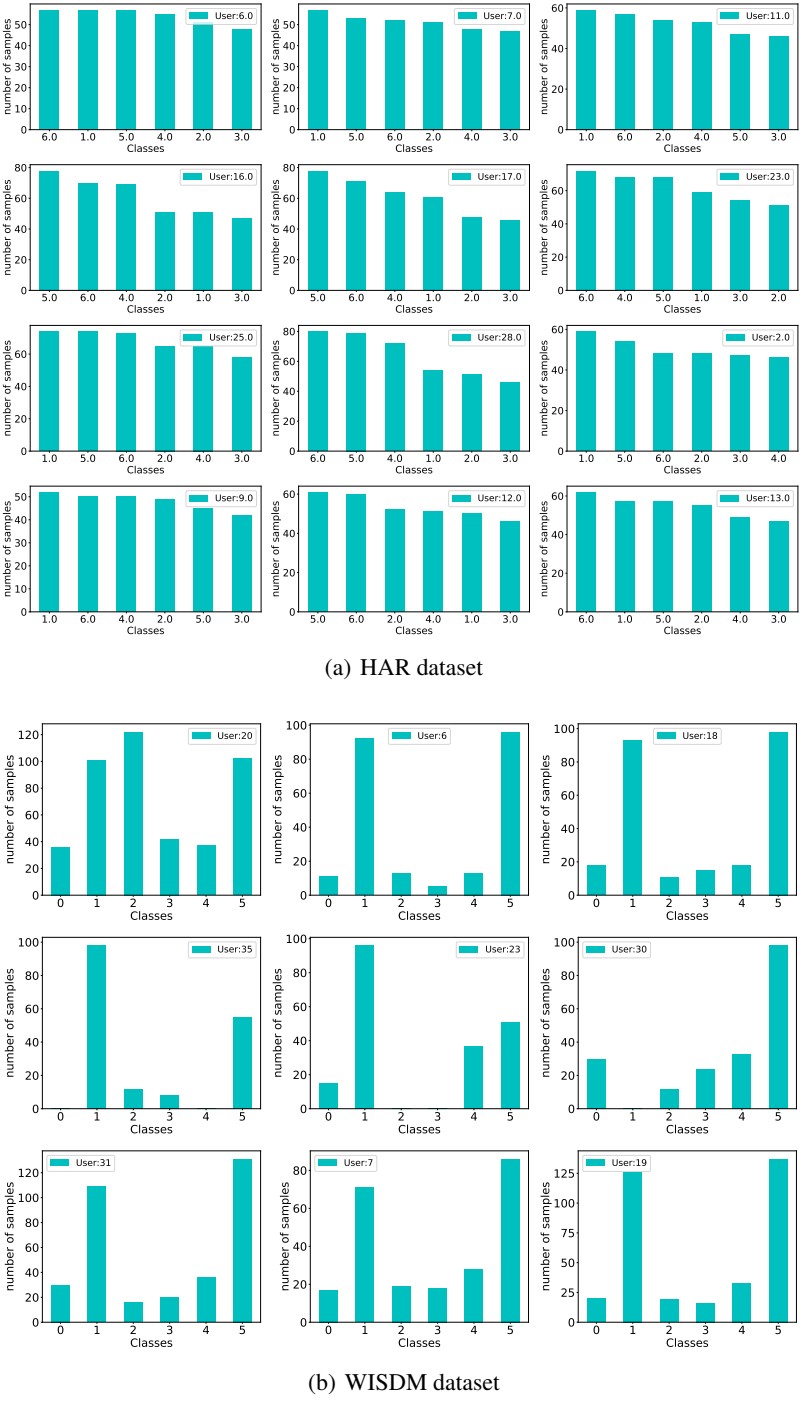

(a) HAR dataset

(b) WISDM dataset

Figure 4: Class distribution of selected subjects among different datasets

## C  HYPER-PARAMETER IMPORTANCE

The importance of each hyper-parameter can be valuable when you have low budget for hyper-parameter tuning. As such you can tune the most important hyper-parameter while fixing others to specific value. In this work, We also study how different hyper-parameters can affect the model performance. We test the learning rate against other model specific performance for 5 different domain adaptation algorithms. To do so, we leverage random forest model and feed the corresponding hyper-parameters as input and the target metric as output (Probst et al., 2019). In our case, we averaged all the model selection risks and use them as a metric to calculate the importance of each parameter.

Figure 5, we study the effect of learning rate, as well as the weights of domain alignment loss (differs according to each method) and the source classification loss. We calculate the importance of these parameters while running the sweeps of three different methods i.e., DDC, Deep-CORAL and AdvSKM. We find that the learning rate is the most significant parameter especially with SSC dataset, as it contributes with more than 80% of the performance. We conclude that more effort should be put in finding the best learning rate that suites each dataset. In addition, we find that the source classification loss comes next in the importance, and hence, more weight should be assigned to it.

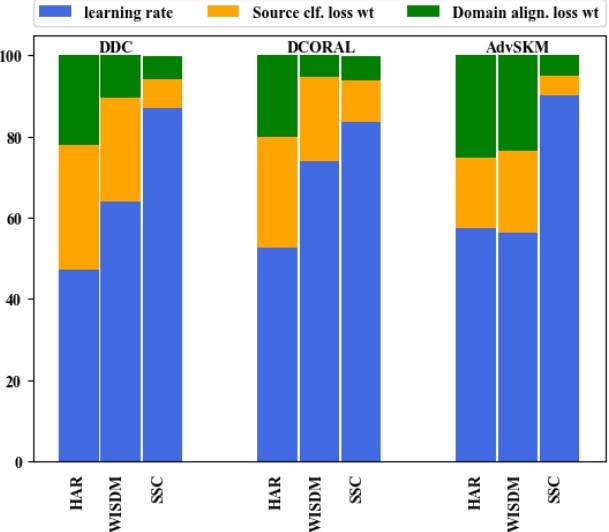

Figure 5: Parameters importance for some selected UDA methods through the three datasets.

# D  DETAILED PARAMETER RANGES FOR THE HYPER-PARAMETER SEARCH

Here, we provide the detailed ranges for each parameter among all selected domain adaptation methods.

Table 8: Details of hyper-parameter tuning setup.

| Method | Hyperparameter | Range |
|---|---|---|
| | Learning Rate | $10^{-2}$ to $10^1$ |
| DDC | MMD loss | $10^{-2}$ to $10^1$ |
| | Classification loss | $10^{-1}$ to $10^1$ |
| Deep CORAL | Coral loss | $10^{-2}$ to $10^1$ |
| | Classification loss | $10^{-1}$ to $10^1$ |
| HoMM | High-order-MMD loss | $10^{-2}$ to $10^1$ |
| | Classification loss | $10^{-1}$ to $10^1$ |
| MMDA | MMD loss | $10^{-2}$ to $10^1$ |
| | Coral Loss | $10^{-2}$ to $10^1$ |
| | Conditional loss | $10^{-2}$ to $10^1$ |
| | Classification loss | $10^{-1}$ to $10^1$ |
| DSAN | Local MMD loss | $10^{-2}$ to $10^1$ |
| | Classification loss | $10^{-2}$ to $10^1$ |
| DANN | MMD loss | $10^{-2}$ to $10^1$ |
| | Classification loss | $10^{-1}$ to $10^1$ |
| CDAN | Adversarial loss | $10^{-2}$ to $10^1$ |
| | Conditional loss | $10^{-2}$ to $10^1$ |
| | classification loss | $10^{-1}$ to $10^1$ |
| DIRT-T | Adversarial loss | $10^{-2}$ to $10^1$ |
| | Conditional loss | $10^{-2}$ to $10^1$ |
| | virtual adversarial | $10^{-2}$ to $10^1$ |
| | Discriminator steps | $10^{-2}$ to $10^1$ |
| | classification loss | $10^{-1}$ to $10^1$ |
| CODATS | Adversarial loss | $10^{-2}$ to $10^1$ |
| | classification loss $\alpha$ | $10^{-1}$ to $10^1$ |
| AdvSKM | Adversarial MMD loss | $10^{-2}$ to $10^1$ |
| | Classification loss | $10^{-1}$ to $10^1$ |

## E    BACKBONE NETWORK ARCHITECTURE

Here, we present the detailed architecture of ADATIME backbone network.

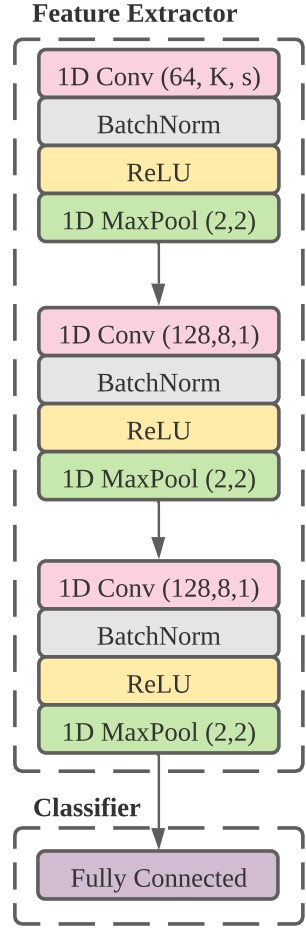

Figure 6: Backbone network of our ADATIME, where $K$ is the kernel size and $s$ is the stride.

## F    PERFORMANCE STUDY OF TWO DIFFERENT BACKBONE NETWORKS

We added 1D-ResNet18 [1] as an additional backbone network. We conducted the experiments on on both small- and large-scale datasets. Table below shows the results of 1D-CNN and 1D-Resnet18 on both HAR and HHAR datasets. We found that our conclusions are consistent among different backbone networks and datasets. Altgouh the absolute performance can be different, the relative performance between different UDA methods is preserved.

Table 9: Evaluation Performance of two backbone network on two different datasets

| Dataset | Networks | DDC | DCoral | HoMM | MMDA | DSAN | DANN | CDAN | DIRT-T | CoDATS | AdvSKM |
|---------|----------|-----|--------|------|------|------|------|------|--------|--------|--------|
| HHAR | CNN | 69.87 | 72.28 | 73.47 | 77.04 | 81.14 | 76.42 | 78.09 | 80.04 | 76.09 | 69.93 |
|  | ResNet18 | 58.426 | 62.32 | 62.69 | 65.31 | 75.08 | 64.97 | 70.75 | 80.24 | 64.58 | 58.67 |
| HAR | CNN | 72.71 | 73.26 | 83.03 | 86.81 | 87.63 | 82.43 | 83.68 | 89.07 | 82.83 | 75.82 |
|  | ResNet18 | 70.59 | 74.53 | 79.1 | 78.19 | 79.71 | 74.05 | 79.13 | 80.745 | 77.75 | 71.77 |

