# OpenReview forum: "A Systematic Evaluation of Domain Adaptation Algorithms On Time Series Data"
_ICLR.cc/2022/Conference — ICLR 2022 Submitted_

### Official Review · Reviewer_S7S5 · 2021-10-30

**Correctness:** 4
**Technical Novelty And Significance:** 2
**Empirical Novelty And Significance:** 2
**Recommendation:** 5
**Confidence:** 5

**Main Review:**

Strengths: A systematic experimental approach  for time series domain adaptation using multiple datasets and baseline architectures.  Comparisons between visual domain adaptation methods and time series-unsupervised domain adaptation methods are given.  Guidelines for future research are given.

**Summary Of The Paper:**

This paper presents an empirical approach for unsupervised domain adaptation of time series data. The paper points out some of the drawbacks of existing approaches due to inconsistencies in evaluation schemes, datasets, model selection rules, and base neural network architectures. The paper then presents adaptations of visual domain adaptation methods for time series data. Experimental results using ten state-of-the-art methods on three benchmark datasets spanning fifteen cross-domain scenarios are presented.

**Summary Of The Review:**

While the authors' efforts in systematic experimental evaluation of existing visual domain adaptation algorithms and time series-unsupervised domain adaptation algorithms is commendable, there is no theoretical understanding at even the most basic level. hence the conclusions drawn from the experiments may be specific to datasets and baseline architectures used. The conclusions do not generalize.

---

> ### Author Response · Authors · 2021-11-20
> **Response to Reviewer S7S5**
>
> We thank the reviewer for the insightful comment. Please find below our response:
>
> > there is no theoretical understanding at even the most basic level. hence the conclusions drawn from the experiments may be specific to datasets and baseline architectures used. The conclusions do not generalize.
>
> The datasets we included in our evaluation are widely adopted and considered as benchmarks for TS-UDA methods. Hence, even if our conclusions were specific to these datasets, our conclusions and findings are still beneficial for current and future research in this area. Nonetheless, we have added the results of an additional HHAR dataset as suggested by reviewers and found our conclusions to hold on this new dataset. This gives us more confidence of the generalizability of our findings.

---

### Official Review · Reviewer_uoXZ · 2021-11-04

**Correctness:** 3
**Technical Novelty And Significance:** 3
**Empirical Novelty And Significance:** 3
**Recommendation:** 5
**Confidence:** 3

**Main Review:**

“Deep learning has achieved a great success in time series classification tasks, assuming access to a vast amount of labeled data for training” I am not totally convinced that this is true. The improvements over simple nearest neighbor models etc is not great.
In the related domain of anomaly detection, a recent paper makes a forceful claim that most of the apparent success of Deep learning for time series anomaly detection is nonsense [a]

The paper contains a nice “bake off”, but the main finding “Visual UDA methods achieve comparable performance to TS-UDA methods on time series data.” is unsurprisingly.

I am very curious about how well one could do with a much simpler methods.
For HAR the class are WALKING, WALKING_UPSTAIRS, WALKING_DOWNSTAIRS, SITTING, STANDING, LAYING. It is easy to tell LAYING from any of the dynamic classes, simply by the fact that the variance of LAYING is less than one tenth of the dynamic classes. It is easy to tell LAYING from the other classes because G (the acceleration due to gravity), sifts from one axis to another etc.
I understand that simply maximizing accuracy is not the full point of the paper, but I am still curious of we really need deep learning here.


“In addition, we find that model selection plays a key role and different selection strategies can significantly affect performance.” It would be surprising of that was NOT true.

However, I do think the experiments as detailed and forceful, and the community may find them useful


[a] Renjie Wu, Current Time Series Anomaly Detection Benchmarks are Flawed and are Creating the Illusion of Progress.


**Summary Of The Paper:**

The paper introduces , a standardized framework to systematically and fairly evaluate different domain adaptation methods on time series data.

**Summary Of The Review:**

While the novelty is low, the experiments as detailed and forceful, and the community may find them useful

---

> ### Author Response · Authors · 2021-11-20
> **Response to Reviewer uoXZ**
>
> Thanks to the reviewer for the insightful comments. Please find our responses below:
>
> > Deep learning has achieved a great success in time series classification tasks, assuming access to a vast amount of labeled data for training” I am not totally convinced that this is true
>
> We toned down our statement in the revision to: "Recently, deep learning is emerging as a promising approach for time series classification". Our initial stronger statement was motivated by the existing literature on deep learning for time series classification [1].
>
> [1] Fawaz, Hassan Ismail, et al. "Deep learning for time series classification: a review." Data mining and knowledge discovery 2019.
>
>
>
> > the main finding "Visual UDA methods achieve comparable performance to TS-UDA methods on time series data." is unsurprisingly.
>
> Our finding contradicts claims in existing TS-UDA papers that visual UDA methods perform poorly on time series data. For example, as quoted from [2]: "However, when met with the unsupervised time series
> domain adaptation (UTSDA) task, a challenging and emerging area of UDA, these methods cannot be robustly applied". In contrast, we have shown experimentally that under a fair and realistic evaluation scheme, methods proposed for visual tasks are competitive with and can even outperform TS-UDA methods.
>
> [2] Liu, Qiao, and Hui Xue. "Adversarial Spectral Kernel Matching for Unsupervised Time Series Domain Adaptation." IJCAI 2021.
>
>
> >  but I am still curious of we really need deep learning here.
>
> Disputing the efficacy of deep learning in time series classification/anomaly detection is out of our scope. Our main scope is to highlight the flaws and inconsistent evaluation schemes of *existing* TS-UDA works for time series data. Yet, it is worth pointing out that a vast amount of literature is leveraging deep learning for time series classification [1,2].
>
> > "In addition, we find that model selection plays a key role and different selection strategies can significantly affect performance." It would be surprising of that was NOT true.
>
> While the importance of model selection may be considered well-known to the reviewer, it is a fact that
> many existing UDA works either do not state the model selection approach, or use a non-realistic approach that depends on target domain labels, violating the main assumption of UDA (see Table 1 in the main paper).

---

### Official Review · Reviewer_f7Xp · 2021-11-04

**Correctness:** 2
**Technical Novelty And Significance:** 1
**Empirical Novelty And Significance:** 2
**Recommendation:** 5
**Confidence:** 4

**Main Review:**

## Strengths
(1) This paper evaluates the previous domain adaption algorithms under a fair setting. Extensive experiments are provided. Also, the experiment results present some competitive baselines to the TS-UDA area. This benchmark will be helpful to future research.

(2) Based on the experiment results, this paper provides some findings or analyses.

(3) The paper is well organized and with clear clarification.

## Weaknesses
(1) The main concern is the technological novelty of this paper. This paper adopts previous methods in the time series datasets, which is technologically trivial. Especially, the proposed AdaTime framework is also trivial. Can you further clarify the difference between your AdaTime framework and the standard domain adaption protocol in image classification? It is hard for me to distinguish them. That is why I think the AdaTime is not novel. More elaborations will be very helpful for my judgment.

(2) Some of the findings are also fragile.
- Performance Gap diminishes with a sufficient amount of data: this clarification is not persuasive for me. Your conclusion is obtained from the comparison among three different datasets. It would be helpful if you controlled the variables. For example, you can keep enlarging the size of one fixed dataset and record the change of results.
- Model selection has a significant effect on performance. I think your experiment results are also affected by the limited data. The three experiment datasets are too small to provide a robust result. Larger datasets should be included, such as the HHAR dataset used in CoDATS.

(3) I am not sure about the contribution of this paper when the backbone is just CNN. Firstly, as shown in Figure 3 of the main text, different backbones result in quite different results. Not only the numerical values but the relative performance is also changed. So I think the findings of your paper may also be changed under other base models. Secondly, as your mentioned, the time series area does not have a consistent backbone. Thus, the experiments are lacking in the backbone aspect. In conclusion, I think more baselines are needed if you want to obtain a general conclusion, such as the TCN.

(4) As for the metric, F1-score is a well-established convention in the time series area, such as anomaly detection and recommended system. Maybe F1-score is not popular in vision, but I still think the metric finding is too trivial.

**Summary Of The Paper:**

This paper explores the unsupervised domain adaptation of time series data (TS-UDA). And it focuses on the benchmark construction. By standardizing the base model, datasets, and model selection, this paper provides a good benchmark of TS-UDA. This benchmark can facilitate future research. Also, the paper proposes some findings.

**Summary Of The Review:**


This paper provides a useful benchmark for the TS-UDA task. Extensive experiments are included. But because of the concern of technology novelty and the unconvincing evidence of some findings, I would like to reject it.

--- after discussion---

The author addressed part of my question. But I still think this paper is below the expectation. I would like to arise the rank to 5.

---

> ### Author Response · Authors · 2021-11-20
> **Response to Reviewer f7Xp (Part 1/2)**
>
> We thank the reviewer for his constructive comments:
> > The main concern is the technological novelty of this paper. This paper adopts previous methods in the time series datasets, which is technologically trivial.
>
> Our goal is not to design a new UDA method but to systematically and fairly benchmark existing UDA methods on time series data, and to do so using a benchmarking methodology that addresses the experimental flaws and inconsistencies in previous works. In this process, we develop a set of best practices that will enable fair and realistic evaluation in future works. To the best of our knowledge, this is the first work to benchmark different UDA methods on time series data.
>
>
> > Some of the findings are also fragile. Performance Gap diminishes with a sufficient amount of data: It would be helpful if you controlled the variables. For example, you can keep enlarging the size of one fixed dataset and record the change of results.
>
> Following the reviewer's suggestion, we conducted additional experiments on two large-scale datasets (i.e., SSC and HHAR), where we vary the training set size for each of the datasets. The results are shown in the table below. The results show that the training set size has no significant effect on the performance gap between different methods. Therefore, thanks to the reviewer's thoughtful comment, we have revised this claim in our paper.
>
> | Dataset | Data Size | DDC   | DCoral | HoMM  | MMDA  | DSAN  | DANN  | CDAN  | DIRT-T | CoDATS | AdvSKM |
> |---------|-----------|-------|--------|-------|-------|-------|-------|-------|--------|--------|--------|
> | HHAR    | 10%       | 60.86 | 63.47  | 63.32 | 70.00 | 59.08 | 69.64 | 69.46 | 72.84  | 64.01  | 60.48  |
> |         | 50%       | 63.93 | 66.81  | 66.46 | 65.14 | 77.40 | 75.18 | 74.09 | 75.66  | 71.87  | 63.37  |
> |         | 100%      | 69.87 | 72.28  | 73.47 | 77.04 | 81.14 | 76.42 | 78.09 | 80.04  | 76.09  | 69.93  |
> | SSC     | 10%       | 50.72 | 50.77  | 49.31 | 49.76 | 54.22 | 50.69 | 47.81 | 54.52  | 54.86  | 52.31  |
> |         | 50%       | 55.85 | 56.13  | 55.95 | 58.22 | 58.89 | 56.04 | 54.59 | 58.03  | 56.44  | 57.72  |
> |         | 100%      | 59.22 | 59.12  | 59.06 | 62.79 | 60.57 | 60.26 | 59.04 | 59.04  | 58.44  | 59.93  |
>
>
>
> > Model selection has a significant effect on performance. I think your experiment results are also affected by the limited data.
>
> While HAR and WISDM are relatively small datasets, we also experimented on the SSC dataset that has considerable number of samples. In addition, following reviewer suggestions, we also added the large scale HHAR dataset to get more conclusive results. The experimental results on HHAR confirmed that model selection strategies can yield different performance for the same UDA algorithm. This reflect the contribution of model selection approach to the performance even on large scale datasets.
>
> To further show the effect of model selection to the performance, we ranked the different UDA methods according to each model selection strategy on HHAR dataset, presented in the below table. We notice that this ranking vary according to each model selection strategy.
>
> |            | SRC Risk | DEV Risk | FST Risk | TGT Risk |
> |------------|:--------:|:--------:|:--------:|:--------:|
> | Deep-Coral |     7    |     8    |     7    |     8    |
> | HoMM       |     6    |     6    |     9    |     7    |
> | MMDA       |     9    |     7    |     6    |     4    |
> | DSAN       |     3    |     1    |     2    |     1    |
> | DANN       |     2    |     5    |     5    |     5    |
> | CDAN       |     1    |     3    |     3    |     3    |
> | DIRT-T     |     4    |     2    |     1    |     2    |
> | CoDATS     |     5    |     4    |     4    |     6    |
> | ADVSKM     |     8    |     9    |     8    |     9    |

---

> > ### Author Response · Authors · 2021-11-20
> > **Response to Reviewer f7Xp (Part 2/2)**
> >
> > > I think more baselines are needed if you want to obtain a general conclusion, such as TCN
> >
> > First, it worth noting that the 1D-CNN is the most adopted backbone network for TS-UDA approaches [1,2], while there is no existing TS-UDA approaches that use TCN as backbone network. However, we are working on the implementation of TCN, and hoping to get the results by the end of the revision period. Currently, to further support our claim, we added 1D-ResNet18 as an additional backbone network. We conducted the experiments on one small and large-scale dataset. We found that our conclusions are consistent among different backbone networks and datasets. The updated results of these experiments are shown in the Table below.
> >
> > | Dataset | Networks | DDC    | DCoral | HoMM  | MMDA  | DSAN  | DANN  | CDAN  | DIRT-T | CoDATS | AdvSKM |
> > |---------|:--------:|--------|--------|-------|-------|-------|-------|-------|--------|--------|--------|
> > | HHAR    |    CNN   | 69.87  | 72.28  | 73.47 | 77.04 | 81.14 | 76.42 | 78.09 | 80.04  | 76.09  | 69.93  |
> > |         | RESNET18 | 58.426 | 62.32  | 62.69 | 65.31 | 75.08 | 64.97 | 70.75 | 80.24  | 64.58  | 58.67  |
> > | HAR     |    CNN   | 72.71  | 73.26  | 83.03 | 86.81 | 87.63 | 82.43 | 83.68 | 89.07  | 82.83  | 75.82  |
> > |         | RESNET18 | 70.59  | 74.53  | 79.1  | 78.19 | 79.71 | 74.05 | 79.13 | 80.745 | 77.75  | 71.77  |
> >
> > [1] Wilson, Garrett, et al. "Multi-Source Deep Domain Adaptation with Weak Supervision for Time-Series Sensor Data." ACM SIGKDD 2020.
> >
> > [2] Liu, Qiao, and Hui Xue. "Adversarial Spectral Kernel Matching for Unsupervised Time Series Domain Adaptation." IJCAI 2021.
> >
> >
> >
> > > Not only the numerical values but the relative performance is also changed.
> >
> > We found that the change in relative performance was mainly due to varying the hyper-parameters of the same UDA methods among different backbone networks. We addressed this issue and unified the hyper-parameters for the same UDA method. The above table shows the results of backbone networks among small and large scale datasets. While the absolute performance is different, the relative performance is consistent among the two backbone networks on both HAR and HHAR datasets. The updated figure can be found in Figure 3 in the modified manuscript.
> >
> > > As for the metric, F1-score is a well-established convention in the time series area, such as anomaly detection and recommended system. Maybe F1-score is not popular in vision, but I still think the metric finding is too trivial.
> >
> > Although F1-score is a well-established metric, most of existing TS-UDA works use the accuracy metric, even with imbalanced datasets, which can be misleading. Hence, we emphasize this finding to highlight this issue to the community.

---

> > > ### Comment · Reviewer_f7Xp · 2021-11-25
> > > **Thanks for your response**
> > >
> > > Many thanks for your response. Your experiments have addressed some of your concerns. But I still have some issues.
> > >
> > > (1) Regarding "technological novelty”, I am not requesting a new algorithm. I need a clear clarification about the difference between your AdaTime framework and the standard domain adaption protocol in image classification.
> > >
> > > Recall the original review as follows:
> > > > Can you further clarify the difference between your AdaTime framework and the standard domain adaption protocol in image classification? It is hard for me to distinguish them. That is why I think the AdaTime is not novel. More elaborations will be very helpful for my judgment.
> > >
> > > (2) Regarding the “F1-score metric”, I still can not admit the contribution of the metric as large progress. If you want to prove this, you can add more insightful experiments. For example, you need to show that one or several methods perform well in precision, but it is extremely poor in F1-score. By doing the above experiments, you can claim the effectiveness of the F1-score in method selection.

---

> > > > ### Author Response · Authors · 2021-11-26
> > > > **Second response to Reviewer  f7Xp**
> > > >
> > > > Many thanks to the reviewer for his response.
> > > >
> > > > >Regarding "technological novelty”, I am not requesting a new algorithm. I need a clear clarification about the difference between your AdaTime framework and the standard domain adaption protocol in image classification.
> > > >
> > > > We thank the reviewer for the clarification.
> > > >
> > > > There are two key differences between our framework and a standard domain adaptation framework:
> > > >
> > > > 1. Standard domain adaptation frameworks do not include the model selection step.
> > > >
> > > > 2. Standard domain adaptation frameworks for image data usually exclude the data preprocessing step. This is because preprocessing steps for benchmark image datasets are standardized, but this may not be the case for time-series datasets.
> > > >
> > > > We elaborate on these points below.
> > > >
> > > > **Model selection:** First, standard domain adaptation frameworks do not include the model selection step. As a result, many previous works fix model hyper-parameters without describing how these are selected, while other works use target labels to select hyper-parameters, violating the main assumption of UDA that no target labels are available.
> > > >
> > > > As can be seen from our results, the model selection procedure used has a significant effect on the performance of the overall domain adaptation procedure. As such, our framework considers the model selection step an integral part of the domain adaptation procedure, and includes multiple practical model selection approaches that are compatible with the assumptions of the domain adaptation setting.
> > > >
> > > > **Preprocessing:** Second, standard domain adaptation frameworks usually exclude the data preparation step from the domain adaptation procedure due to the standardization of preprocessing steps for existing benchmark image datasets. In contrast, for time series applications, there is no consistent preprocessing even for the same dataset. For instance, while slicing the long time series into short windows is a key step in time series preprocessing, different window sizes can be used which can significantly affect performance. Hence, to address this issue, we include a standardized data preparation step when comparing different UDA methods to ensure fair evaluation.
> > > >
> > > > > Regarding the “F1-score metric”, I still can not admit the contribution of the metric as large progress. If you want to prove this, you can add more insightful experiments. For example, you need to show that one or several methods perform well in precision, but it is extremely poor in F1-score. By doing the above experiments, you can claim the effectiveness of the F1-score in method selection.
> > > >
> > > > We would like to point the reviewer to the results from a similar experiment reported in Figure 2 of the main paper, which shows that some methods perform well in terms of accuracy but perform extremely poorly in terms of F1-score. To further support this finding, the table below shows the performance on both WISDM and SSC  datasets in terms of F1-score and accuracy and their respective ranking. For WISDM dataset, while deep CORAL (DCORAL) and CDAN perform well in terms of accuracy, ranking 2nd and 3rd, they show extremely poor performance in terms of F1-Score where their ranking drops significantly to 8th and 6th respectively. Similarly, for SSC dataset, while DANN shows very good performance in terms of accuracy by ranking 3rd, its performance drops significantly in terms of F1-score and its ranking goes down to 8th. To sum up, we have clearly shown for two selected class-imbalanced datasets (i.e., WISDM and SSC) that multiple methods can perform well in terms of accuracy but perform very poor in terms of F1-score. We are also ready to add more experiments upon the reviewer request to further address his concern.
> > > >
> > > >
> > > > |   |       |     WISDM    |          |         |  |       |   SSC         |         |
> > > > |:------:|:-----:|:-----:|:--------:|:-------:|:-----:|:-----:|:--------:|:-------:|
> > > > | Method |  ACC  |   F1  | Rank_ACC | Rank_F1 |  ACC  |   F1  | Rank_ACC | Rank_F1 |
> > > > |  DDC  | 74.79 | 55.03 |     5    |    10   | 69.99 | 59.22 |     7    |    5    |
> > > > | DCORAL | 75.73 | 57.43 |     2    |    8    | 69.82 | 59.12 |    10    |    6    |
> > > > |  CDAN  | 75.35 | 57.85 |     3    |    6    | 69.93 | 59.06 |     9    |    7    |
> > > > | CoDATS | 73.69 | 56.75 |     6    |    9    | 73.38 | 62.79 |     1    |    1    |
> > > > |  HoMM  | 74.82 | 62.98 |     4    |    3    | 71.82 | 60.57 |     2    |    2    |
> > > > |  MMDA  | 72.87 | 63.97 |     8    |    2    | 70.95 | 60.26 |     6    |    3    |
> > > > |  DSAN  | 69.06 | 60.07 |     9    |    5    | 71.27 | 59.04 |     5    |    8    |
> > > > |  DANN  |   68  | 57.81 |    10    |    7    | 71.76 | 59.04 |     3    |    8    |
> > > > |  DIRT  | 78.74 | 66.27 |     1    |    1    | 69.96 | 58.44 |     8    |    10   |
> > > > | ADVSKM | 73.55 | 60.54 |     7    |    4    | 71.52 | 59.93 |     4    |    4    |

---

> > > > > ### Comment · Reviewer_f7Xp · 2021-11-29
> > > > > **Thanks for your response**
> > > > >
> > > > > Many thanks to the authors for the detailed explanation and more results.
> > > > >
> > > > > The AdaTime is more clear to me. But I still can not appreciate the work a lot because of the following issues:
> > > > > - The contribution of model selection. I think the model selection has been noticed as a necessary part of the domain adaption framework since the methods that you used in AdaTime are from the image domain adaption.
> > > > > - The contribution of preprocessing. I appreciate the contribution in this aspect, which is necessary for a benchmark. But I think, as a benchmark, you can do more about just comparison. For example, there are more pre-processing methods, which one is better? This can benefit the community.
> > > > >
> > > > > Thanks for the results of F1 score. As shown in your table, if I want to select the best algorithm (Top 1), the accuracy metric is still effective. But I slightly accept the contribution of introducing F1-score to classification.
> > > > >
> > > > > This paper provides detailed comparisons but still lacks insights, practical findings. For example,
> > > > > - Why the Visual UDA methods are useful for time series? The visualization of the learned feature space is expected.
> > > > > - Since the backbone is pivotal, which one should I use? What about nowadays Transformer-based backbones?
> > > > >
> > > > > Thus, I would like to raise the rank to 5. More explanations are accepted if the author holds opposite ideas.

---

> > > > > > ### Author Response · Authors · 2021-11-30
> > > > > > **Response to Reviewer f7Xp**
> > > > > >
> > > > > >
> > > > > > We are really thankful for the reviewer's response and for his constructive comments.
> > > > > >
> > > > > > > The contribution of model selection. I think the model selection has been noticed as a necessary part of the domain adaption framework since the methods that you used in AdaTime are from the image domain adaption.
> > > > > >
> > > > > > Our work includes two UDA methods proposed for time series data, and the conclusion that model selection is important also holds for these. Therefore, the finding of the contribution of model selection is not specific to image domain adaptation methods but also generalizes to methods proposed for time series.
> > > > > >
> > > > > > Moreover, apart from highlighting the critical role of model selection, another aim of our work is to address the use of inappropriate model selection approaches that do not align with UDA assumptions. Specifically, many previous works utilize target domain labels (TGT) to select their hyper-parameters (see Table 1 in the modified paper), which, while providing an upper bound on achievable performance, fundamentally violates the assumption in unsupervised domain adaptation that no labeled samples are available in the target domain. Therefore, in addition to TGT, we evaluated several other realistic model selection approaches to select the hyper-parameters, and have provided insights on which ones to use in Section 3.3.
> > > > > >
> > > > > >
> > > > > > > The contribution of preprocessing. I appreciate the contribution in this aspect, which is necessary for a benchmark. But I think, as a benchmark, you can do more about just comparison. For example, there are more pre-processing methods, which one is better? This can benefit the community.
> > > > > >
> > > > > > The main goal of our approach is to promote fair comparison between different UDA works and ensure that the performance improvement is only due to the UDA method, which is the core part of any UDA method. Therefore, We included the unified preprocessing in our framework to remove the contribution of different preprocessing techniques to the performance, ensuring that performance improvement is only due to the proposed UDA method. Nevertheless, following the reviewer's suggestion, we aim to include a study of different preprocessing techniques to further benefit the community and future research in this area.
> > > > > >
> > > > > >
> > > > > > > Why the Visual UDA methods are useful for time series? The visualization of the learned feature space is expected.
> > > > > >
> > > > > > The core domain adaptation components in many existing Visual UDA methods are applied to the feature space, which means they can be applied to different data modalities once an appropriate feature representation is generated. Generating such an appropriate feature representation is typically achieved through the choice of a backbone network for specific data modalities. Hence, with the appropriate backbone network that can encode the time series raw signal into vectorized features, the domain adaptation component of the Visual UDA methods should perform well and can be strong baselines for TS-UDA methods. Therefore, TS-UDA methods should compare against Visual UDA methods rather than claiming that they are not working properly. Nevertheless, following the reviewer's suggestion, we will add the feature visualization in a further revision to provide more intuition about the success of Visual UDA methods.
> > > > > >
> > > > > > > Since the backbone is pivotal, which one should I use? What about nowadays Transformer-based backbones?
> > > > > >
> > > > > > Although it is clear that the backbone network is pivotal, different backbone networks have still been used in previous works when comparing between UDA methods. For instance, some methods that use recurrent neural network as the backbone network are compared against methods with convolutional-based backbone networks. Therefore, to address this issue, we aimed to show that the backbone network has a significant influence on the performance and should be kept fixed to fairly compare between different UDA methods. Nevertheless, following the reviewer's suggestion, in a further revision, we will study the generalization ability of different backbone networks, including recent backbones, and recommend the best backbone network for UDA on time series data.

---

### Official Review · Reviewer_oysD · 2021-11-05

**Correctness:** 3
**Technical Novelty And Significance:** 3
**Empirical Novelty And Significance:** 3
**Recommendation:** 5
**Confidence:** 4

**Main Review:**

Pros:
1. The paper comprehensively adapts the state-of-the-art UDA methods to time series classification tasks.
2. The paper proposes a novel UDA framework for time series data, which may contribute to the community.
3. The idea of this paper is novel to some extent.

Cons:
1. My main concern is the motivation. The paper only talks about how the framework is designed, while not elaborating clearly on why the framework is designed like this.
2. The selected datasets are relatively too small and simple that the backbone network can only be a 1D-CNN in order to avoid the overfitting phenomenon. The experimental results on these toy time-series datasets are unconvincing.
3. The organization and writing of this paper should be improved. The authors should pay more attention to the motivation and the details of the framework.
4. It would be better if the authors explain some symbols like $X_{train}^{S}$, $X_{test}^{S}$, $Z_{train}^{S}$, $Z_{test}^{S}$, ..., etc. in Figure
5. Some related works are missing:

[1] Ruichu Cai, Jiawei Chen, Zijian Li, Wei Chen, Keli Zhang, Junjian Ye, Zhuozhang Li, Xiaoyan Yang, Zhenjie Zhang:
Time Series Domain Adaptation via Sparse Associative Structure Alignment. AAAI 2021
[2] Xiaoyong Jin, Youngsuk Park, Danielle C. Maddix, Yuyang Wang, Xifeng Yan. Domain Adaptation for Time Series Forecasting via Attention Sharing. arXiv:2102.06828

Minors:
On page 2, “the the following questions" -> "the following questions"

**Summary Of The Paper:**

The paper proposes a systematic evaluation framework named ADATIME, which systematically evaluates different unsupervised domain adaptation methods on time series data. The whole framework consists of a feature extractor, a classifier, and a domain alignment component. The paper conducts large-scale experiments adapting the state-of-the-art visual domain adaptation methods to the proposed framework on time series classification tasks. The findings based on the experimental results reveal the key points of applying UDA to time series data.

**Summary Of The Review:**

The findings are interesting and may contribute to the community. However, considering the motivation and the unconvincing experiments, I vote for "5: marginally below the acceptance threshold".

---

> ### Author Response · Authors · 2021-11-20
> **Response to Reviewer oysD**
>
> We thank the reviewer for his constructive comments. Please find our answers below:
>
> > The paper only talks about how the framework is designed, while not elaborating clearly on why the framework is designed like this.
>
> Existing TS-UDA works suffer from inconsistent evaluation schemes, datasets, and backbone networks. These issues can significantly affect experimental results causing improved performance to be misattributed to the proposed domain adaptation component, when improvements can be simply due to other changes unrelated to the core domain adaptation method (e.g. backbone).  Therefore, we design our benchmarking approach in a way to address these issues. For instance, to remove the effect of different backbone networks, we use the same backbone network when comparing between different UDA methods. The table below summarizes the existing experimental flaws and our corresponding design decision. We also provided further details about the motivation of our work in the introduction section.
>
>
> | Problem                                    | Design Decision                                                        |
> |--------------------------------------------|------------------------------------------------------------------------|
> | Inconsistent backbone networks             | We fixed the same backbone network for all methods                     |
> | Using target labels for model selection    | We provided realistic model selection approaches without target labels |
> | Different training procedures              | We unified the evaluation schemes and training procedure               |
> | Using accuracy metric with imbalanced data | We use F1-score as the main metric                                     |
>
>
>
> > the selected datasets are relatively too small and simple; The experimental results on these toy time-series datasets are unconvincing
>
> The selected datasets are commonly used for benchmarking TS-UDA methods as in [1,2]. Moreover, while performance may be high on these datasets in the i.i.d. setting, performance drops significantly in the cross-domain scenarios that are the focus of this work.
> To demonstrate this gap, we compare the target-only (i.e., training and testing on the same domain) and the source-only (i.e., training and testing on different domains without adaptation) performance on these datasets in the Table below. It is evident there is a large performance gap between the two scenarios, showing the challenge in the cross-domain scenarios of interest even with these datasets.
>
> We have also included a larger HHAR dataset in our experiments and the results add further support for our conclusions.
>
> |                            |   HAR  |  HHAR | WISDM |  SSC  |
> |----------------------------|:------:|:-----:|:-----:|:-----:|
> | Same Domain  (Target-only) | 100.00 | 98.76 | 98.01 | 73.73 |
> | Cross-Domain (Source-only) |  62.68 | 64.90 | 48.57 | 55.35 |
> | Gap ($\delta$)             |  37.32 | 33.86 | 49.44 | 18.38 |
>
> [1] Wilson, Garrett, et al. “Multi-Source Deep Domain Adaptation with Weak Supervision for Time-Series Sensor Data.” ACM SIGKDD 2020.
>
> [2] Liu, Qiao, and Hui Xue. “Adversarial Spectral Kernel Matching for Unsupervised Time Series Domain Adaptation.” IJCAI 2021.
>
>
> > The authors should pay more attention to the motivation and the details of the framework.
>
> We further elaborated the motivation of our approach in our response to the first comment and in the introduction section of the modified manuscript. Besides, we further explained the details of our framework in Section 3.
>
> > It would be better if the authors explain some symbols like $X_{train}^S, X_{test}^S, Z_{train}^S, Z_{test}^S $..., etc. in Figure
>
> We have included all the missing explanations of the symbols in our main Figure for clarity.
>
> > Some related works are missing
>
> Our goal is to address the UDA for time series classification. Therefore, we did not include the mentioned DAF [1] reference, which is mainly proposed for time series forecasting.
>
> Regarding SASA [2], it mainly relies on the LSTM hidden states to align the domains. Hence, their approach is only specific to LSTM network and cannot be applied with CNN-based backbone networks to fairly compare it with other baselines that are generic and can be used with any backbone architecture.
>
> [1] Xiaoyong Jin, et al. "Domain Adaptation for Time Series Forecasting via Attention Sharing", ArXiv 2021.
>
> [2] Ruichu Cai, et al. "Time Series Domain Adaptation via Sparse Associative Structure Alignment", AAAI 2021.

---

### Official Review · Reviewer_X8V1 · 2021-11-06

**Correctness:** 3
**Technical Novelty And Significance:** 1
**Empirical Novelty And Significance:** 2
**Recommendation:** 3
**Confidence:** 3

**Main Review:**

Strength:
- include extensive baseline for the experiment
- open-sourced the code


Weakness:
- did not clarify what is the specific challenge in time series data beyond static data, and how to remedy them as contributions?
- does not cover other main time series applications: time series forecasting scenarios.
- did not cover or compare with the recent work (DAF) of domain adaptation for time series by Jin et al. https://arxiv.org/abs/2102.06828
--  did not justify well why domain alignments are important well. In DAF, it mentioned having both domain specific and invariance features are critical.
-- what if using different backborns, such as attention-based ones?
- hard to understand why many criteria are proposed. what is difference between SCC and DEV risk? when to select one of them?
- difficult to interpret the experiment results
-- what is visual UDA in the table 3?
-- why were average values reported?
- the observations are not much informative
-- one on imbalanced data is not connected to model selection criterions.
-- one of backborn selection is kind of implying non-systemtic aspect. What is the real take-away out of this? may it recommend to use other possible backborn, e.g., attention-based one?
- hard to understand why authors emphasize fair and realistic procedure. Are any of other methods not fair or realistic and why?
-

**Summary Of The Paper:**

This work proposed a systematic framework of (unsupervised) domain adaptation for time series data, with various model selections and strategies.

**Summary Of The Review:**

The paper is not well-written especially experiments parts, and the message of our them are not clear in the line of overall paper take-away.Therefore, the contribution seems to be not clear beyond aggregating all existing DA methods and systematically deploying them.

---

> ### Author Response · Authors · 2021-11-20
> **Response to Reviewer X8V1 (Part 1/2)**
>
>
> We thank the reviewer for the insightful comments. Please find our answers below:
>
> > did not clarify what is the specific challenge in time series data beyond static data, and how to remedy them as contributions?
>
> In general, different from static data, time series data has inherent temporal dynamics property, which has to be carefully considered during the learning process. Moreover, we address a more challenging case of time series classification under the distribution shift problem.
> We updated the paper to further clarify the difficulties associated with time series data.
>
> > does not cover other main time series applications; did not cover or compare with the recent work (DAF) of domain adaptation for time series.
>
> The focus of our work is evaluating domain adaptation methods in the context of time-series classification. While other time-series applications are also interesting to consider, this was not the goal of the paper. As such, we did not include DAF in our benchmarking, as it is a method for time series forecasting not classification.
>
> > did not justify well why domain alignments are important well? In DAF, it mentioned having both domain specific and invariance features are critical; what if using different backborns, such as attention-based ones?
>
> Domain alignment is a general term that refers to the adaption step in the UDA method. The adaptation approach can be domain-invariant or domain-specific according to each UDA method. In this work, we propose a benchmarking methodology to re-evaluate *existing* UDA methods, and our goal is not to propose a new UDA method. Instead, we address the flaws and inconsistencies in evaluations of existing TS-UDA approaches and show the real performance of different TS-UDA methods under fair and realistic evaluation schemes. To address the reviewer concern about backbones, we have included one more backbone network in our experiments, i.e., 1D-ResNet18.
>
> > hard to understand why many criteria are proposed. what is difference between SCC and DEV risk? when to select one of them?
>
> We wish to clarify that the different model selection criteria are not proposed by us, but are rather commonly used in the literature. One of the goals of our work is in fact to answer the reviewer's question about which one to use in a realistic evaluation and we recommend the use of source (SRC) or few-shot target risk (FST).
>
> To elaborate, many previous works utilize target domain labels (TGT) to select their hyper-parameters (see Table 1 in the modified paper), which, while providing an upper bound on achievable performance, fundamentally violates the assumption in unsupervised domain adaptation that no labeled samples are available in the target domain. Therefore, in addition to TGT we evaluated several other realistic model selection approaches to select the hyper-parameters.
>
> For instance, the source risk (SRC) is one approach that can be easily adopted by UDA methods, since it is estimated using only labeled source data. However, the effectiveness of the source risk is mainly controlled by the sample size of source data and severity of distribution shift. Differently, the deep embedded evaluation (DEV) risk aims to address severe distribution shift by considering the relationship between the source and target domains. However, it is more complex and computationally expensive than the source risk. Besides, it can be unstable with smaller sample sizes of source and target domains. Finally, we considered the few-shot target risk (FST), which may be a realistic compromise where a few labeled target domain examples are available for model selection. We have included further discussion about the advantages and disadvantages of each model selection criteria in Section 3.3 to address the reviewer's comments.
>
>
> > what is visual UDA in the table 3?
>
> In Table 3, visual UDA refers to the UDA methods proposed for visual applications (as clarified in Section 1, Section 3.2, and Table 1).
>
> > why were average values reported?
>
> It is not clear whether the reviewer is asking about the Avg/alg column in Table 3, or about reporting the average performance of all cross-domain scenarios for each algorithm, so we will address both questions. Regarding the Avg/alg column, we included it to show the average performance of each algorithm among all the model selection criteria. However, as it seems a bit confusing, we removed it accordingly. On the other hand, we reported the mean results of the cross-domain scenarios in Table 3 due to the space limitations. However, the detailed results for each cross-domain scenario are provided in Tables 4, 5, 6 and 7 in the supplementary materials.

---

> > ### Author Response · Authors · 2021-11-20
> > **Response to Reviewer X8V1 (Part 2/2)**
> >
> > > The observations are not much informative; one on imbalanced data is not connected to model selection criterion.
> >
> > Generally, our scope is to address inconsistencies and flaws of experimental setting of existing TS-UDA works. The usage of accuracy with imbalanced data and the lack of realistic model selection are two main issues in the existing literature of TS-UDA.
> >
> > > one of backborn selection is kind of implying non-systemtic aspect. What is the real take-away out of this?
> >
> > The main message that we aim to deliver is that the backbone network has a significant influence on the performance of the UDA method, independent of the actual domain adaptation component. Hence, it needs to be kept fixed to fairly compare between different UDA methods. Nevertheless, to address the reviewer concern about using a single backbone network, we included two backbone networks, 1D-CNN and 1D-Resnet18 in our experiments, as shown in the table below (the respective figure can be found in Figure 3 in the modified manuscript). Although the absolute performance varies due to different backbone networks, the relative performance between different UDA methods is still consistent.
> >
> > | Dataset | Networks |   DDC  | DCoral |  HoMM |  MMDA |  DSAN |  DANN |  CDAN | DIRT-T | CoDATS | AdvSKM |
> > |:-------:|:--------:|:------:|:------:|:-----:|:-----:|:-----:|:-----:|:-----:|:------:|:------:|:------:|
> > |   HHAR  |    CNN   |  69.87 |  72.28 | 73.47 | 77.04 | 81.14 | 76.42 | 78.09 |  80.04 |  76.09 |  69.93 |
> > |         | ResNet18 | 58.43 |  62.32 | 62.69 | 65.31 | 75.08 | 64.97 | 70.75 |  80.24 |  64.58 |  58.67 |
> > |   HAR   |    CNN   |  72.71 |  73.26 | 83.03 | 86.81 | 87.63 | 82.43 | 83.68 |  89.07 |  82.83 |  75.82 |
> > |         | ResNet18 |  70.59 |  74.53 |  79.1 | 78.19 | 79.71 | 74.05 | 79.13 | 80.75 |  77.75 |  71.77 |
> >
> >
> > > hard to understand why authors emphasize fair and realistic procedure. Are any of other methods not fair or realistic and why?
> >
> > Existing works on time series domain adaptation suffer from inconsistencies in evaluation schemes, datasets,  and backbone networks. For instance, some methods that use recurrent neural network as the backbone network are compared against methods with convolutional based backbone networks [1,2]. Besides, some methods use labeled target data for hyper-parameter selection, which violates the fundamental assumption of unsupervised domain adaptation. In particular, we found that 5 out of the 10 papers describing the methods we evaluate, mention that target domain labels are used to select the hyper-parameters, as shown in Table 1. In addition, the other three papers used fixed hyper-parameters in their experiments, without describing how these were selected. All the aforementioned issues can significantly affect the performance of the method while being mistakenly attributed to the domain adaptation component, which is meant to be the key contribution of the work. To address these issues, our benchmarking methodology removes all extraneous factors to ensure that performance improvement is only due to the proposed UDA algorithm.
> >
> >
> > [1] Sanjay Purushotham, Wilka Carvalho, Tanachat Nilanon, and Yan Liu.  “Variational recurrent adversarial deep domain adaptation.”   In ICLR 2017.
> >
> > [2] Wilson, Garrett, et al. “Multi-Source Deep Domain Adaptation with Weak Supervision for Time-Series Sensor Data.” ACM SIGKDD 2020.

---

### Author Response · Authors · 2021-11-20
**General response to all reviewers**

We would like to thank the reviewers for their thoughtful and constructive comments. We are glad that reviewers agree that our rigorous benchmarking along with our findings contribute to the community, and can be helpful for the future research. Here, we address the shared concerns among reviewers, and then we respond to the comments raised by each reviewer individually. The corresponding modifications are highlighted in blue in the manuscript.

**Novelty**: Our goal is not to design a new UDA method but to systematically and fairly benchmark existing UDA methods on time series data, and to do so using benchmarking methodology that addresses the experimental flaws and inconsistencies in previous works. In this process, we develop a set of best practices that will enable fair and realistic evaluation in future works. To the best of our knowledge, this is the first work to benchmark different UDA methods on time series data.

**Datasets**:  Some reviewers had concerns that the selected datasets are too small and simple to draw conclusions. We have now included the suggested larger HHAR dataset in our experiments and the results add further support for our conclusions. We further wish to highlight that the datasets we used are commonly used for benchmarking TS-UDA methods as in [1,2]. Moreover, while performance may be high on these datasets in the i.i.d. setting, performance drops significantly in the cross-domain scenarios that are the focus of this work.
To demonstrate this gap, we compare the target-only (i.e., training and testing on the same domain) and the source-only (i.e., training and testing on different domains without adaptation) performance on these datasets in the Table below. It is evident there is a large performance gap between the two scenarios, showing the challenge in the cross-domain scenarios of interest even with the small datasets.

|                            |   HAR  |  HHAR | WISDM |  SSC  |
|----------------------------|:------:|:-----:|:-----:|:-----:|
| Same Domain  (Target-only) | 100.00 | 98.76 | 98.01 | 73.73 |
| Cross-Domain (Source-only) |  62.68 | 64.90 | 48.57 | 55.35 |
| Gap ($\delta$)             |  37.32 | 33.86 | 49.44 | 18.38 |

[1] Wilson, Garrett, et al. “Multi-Source Deep Domain Adaptation with Weak Supervision for Time-Series Sensor Data.”, ACM SIGKDD 2020.

[2] Liu, Qiao, and Hui Xue. “Adversarial Spectral Kernel Matching for Unsupervised Time Series Domain Adaptation.”, IJCAI 2021.



**Backbone networks**: Some reviewers expressed concerns that our findings may be specific to a single backbone network. To address this concern, we have included one more backbone network, 1D-Resnet18, in our experiments. Although the absolute performance of each UDA method varies according to the backbone network used, the relative performance among different methods is consistent. This suggests that the rest of our conclusions are likely to generalize to different backbone networks.

### **Revisions to paper:**
We have also revised the paper accordingly to address these and other comments, and we summarize the major changes in the paper as follows:
- Evaluation on additional large scale HHAR dataset (Section 5).
- Evaluation on additional backbone network (Section 5).
- Clarification of the motivation of the proposed approach (Section 1).
- Explanation of the advantages and disadvantages for each model selection strategy (Section 3.3).

### **Contributions:**
Finally, we would like to re-iterate the contributions and impact of this work. There is a growing body of literature applying unsupervised domain adaptation for time series data (i.e., TS-UDA methods). Yet, existing TS-UDA baselines suffer from inconsistent evaluation schemes and backbone networks. Besides, they often utilize target domain labels to select the model parameters, violating the unsupervised assumption of domain adaptation. The aforementioned issues can contribute to the performance, and the performance gain is mistakenly attributed to the proposed UDA method. To address these issues, our benchmarking evaluation suite (ADATIME) uses a standardized evaluation scheme, and more realistic model selection techniques that align with UDA assumptions. The code of ADATIME will be made publicly available for researchers to enable seamless evaluation of different domain adaptation methods on time series data. Our work provides a foundation and best practices for fair and realistic evaluation of current and future TS-UDA methods.

---

### Author Response · Authors · 2021-12-01
**Follow-up**

We thank the reviewers again for their constructive and valuable comments, which have contributed to improving the quality of our manuscript. As the discussion period is closing soon, we would appreciate it if the reviewers can provide any further feedback or follow-up to our first-round response.

We look forward to hearing from you soon.

Thank you

---

### Decision · Program_Chairs · 2022-01-20

**Decision:**

Reject

**Comment:**

This work aims at giving a systematic evaluation of different unsupervised domain adaptation methods on time series classification tasks under a fair setting. By providing extensive experiments on various datasets, competitive baselines, and model selection approaches, this paper has the potential to facilitate future research on this topic if the mentioned concerns are well addressed.

After rebuttal and discussion, the final scores were 3/5/5/5/5. AC considered all reviews, author responses, and the discussions, as well as reading through the paper as a neutral referee, and reject the paper based on the following concerns:
+ *Model Selection Criterion*: As stated by the authors, employing labeled target data for model selection will violate the fundamental assumption of unsupervised domain adaptation. However, the proposed Few-Shot Target Risk (FST Risk) also requires labeling a few target domain samples. If it is possible, why not directly conduct semi-supervised domain adaptation?
+ *Experiment Details*: As a benchmark paper, it is extremely important to carefully design the experiment details to attain promising results. Among these details, a suitable network backbone for time series classification (Is CNN or ResNet-18 the best choice?  Or TCN mentioned by Reviewer f7Xp), large-scale datasets with considerable domain gap, and evaluation metrics are the first consideration to attain insightful findings.
+ *Novelty or Interesting Findings*: As pointed out by reviewers, it is obvious that the technical novelty is limited but it may be okay for a benchmark paper if solid/interesting experimental results are observed. However, some of the findings are also fragile and the experiments should be carefully conducted to make them more solid.

In summary, this paper studies a promising research direction of domain adaptation, but the work cannot be accepted before addressing the reviewers' comments. The weaknesses mentioned above will have a high probability of being asked by the reviewers of the next conference. So the authors need to make sure that they substantially revise their work before submitting it to another venue.